# Taming Hierarchical Image Coding Optimization: A Spectral Regularization Perspective

**Wuyang Cong, Junqi Shi, Ming Lu[†], Xu Zhang, Zhan Ma[†]**
School of Electronic Science and Engineering, Nanjing University
{congwuyang}@smail.nju.edu.cn, {minglu,mazhan}@nju.edu.cn

## Abstract

Hierarchical coding offers distinct advantages for learned image compression by capturing multi-scale representations to support scale-wise modeling and enable flexible quality scalability, making it a promising alternative to single-scale models. However, its practical performance remains limited. Through spectral analysis of training dynamics, we reveal that existing hierarchical image coding approaches suffer from cross-scale energy dispersion and spectral aliasing, resulting in optimization inefficiency and performance bottlenecks. To address this, we propose explicit spectral regularization schemes for hierarchical image coding, consisting of (i) intra-scale frequency regularization, which encourages a smooth low-to-high frequency buildup as scales increase, and (ii) inter-scale similarity regularization, which suppresses spectral aliasing across scales. Both regularizers are applied only during training and impose no overhead at inference. Extensive experiments demonstrate that our method accelerates the training of the vanilla model by $2.3\times$, delivers an average 20.65% rate–distortion gain over the latest VTM-22.0 on public datasets, and outperforms existing single-scale approaches, thereby setting a new state of the art in learned image compression.

## 1 Introduction

Learned image compression (LIC) (Chen et al., 2021; Lu et al., 2021; He et al., 2022; Duan et al., 2023a; Liu et al., 2023; Li et al., 2023; Qin et al., 2024; Fu et al., 2024; Li et al., 2025c; Zeng et al., 2025; Jiang et al., 2025) has recently surpassed traditional hand-crafted codecs (Wallace, 1991; Bellard, 2015; Bross et al., 2021) in compression performance, largely benefiting from statistical learning and end-to-end optimization. Most existing LIC frameworks are built upon single-scale variational autoencoder (VAE) architectures, where the reconstruction relies on a single-scale latent representation and auxiliary variables (e.g., hyperpriors (Ballé et al., 2018)) are primarily used for entropy modeling. Within this paradigm, progress has been driven by powerful transformation networks (Lu et al., 2021; Liu et al., 2023; Qin et al., 2024; Zeng et al., 2025) and advanced context modeling (Ballé et al., 2018; Minnen et al., 2018; Cheng et al., 2020). While this design has proven highly effective, its performance is approaching saturation, particularly in high-bitrate and high-resolution scenarios.

To overcome the performance plateau of single-scale architectures, recent efforts have turned to hierarchical VAE (HVAE) designs (Duan et al., 2023b;a; Lu et al., 2024; Zhang et al., 2025a), which extend the single-scale processing to multiple scales. Such hierarchical representations are in principle well-suited for compression: they provide multi-scale signal descriptions, enable scale-wise autoregressive modeling, and support flexible quality scalability (Wallace, 1991; Schwarz et al., 2007; Boyce et al., 2015). However, their empirical performance has not yet matched these theoretical advantages. For instance, QARV (Duan et al., 2023a), one of the most representative hierarchical schemes, requires nearly 10 days of training on a single NVIDIA RTX 3090 GPU and still underperforms lighter single-scale models such as ELIC (He et al., 2022) in a certain bitrate range. These limitations suggest that the potential of hierarchical coding remains far from fully exploited.

We identify that the challenge lies in the naive optimization approach, which overlooks the intended information allocation across scales. In hierarchical architectures, higher scales are expected to

---

[†]Corresponding authors.

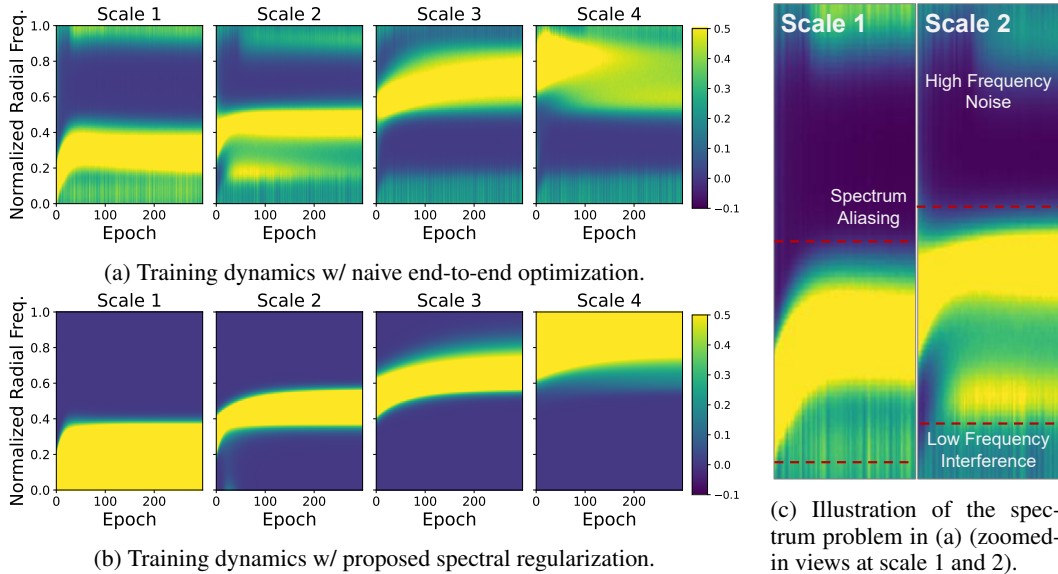

(a) Training dynamics w/ naive end-to-end optimization.

(b) Training dynamics w/ proposed spectral regularization.

(c) Illustration of the spectrum problem in (a) (zoomed-in views at scale 1 and 2).

Figure 1: **Spectral energy dynamics during hierarchical compression training.** (a) naive training—exhibits spectral dispersion, noise, and aliasing issues; (b) regularized training—shows clear and decoupled coarse-to-fine structure; (c) zoomed-in illustration of these spectral issues. More implementation details and plotting scripts can be found in Appendix A.2

capture low-frequency global structures, whereas lower scales should represent high-frequency details (Sønderby et al., 2016; Vahdat & Kautz, 2020). However, naive training fails to respect this information hierarchy: it optimizes all scales over the full frequency spectrum, leading to undesirable outcomes as illustrated in Fig. 1a. Specifically, (i) the energy of each scale becomes dispersed across frequencies, hampering its compact representation and convergence speed; and (ii) different scales exhibit severe spectral aliasing, resulting in the encoding of redundant frequency components. Both factors contribute to the reduced compression performance observed.

This observation naturally raises a question: *Can hierarchical models be explicitly trained toward frequency-stratified representations to fully exploit their potential?* In this context, prior works have suggested a *frequency principle* (Rahaman et al., 2019; Ronen et al., 2019; Xu & Zhou, 2021; Xu et al., 2025), indicating that different network layers exhibit distinct sensitivities to different frequency bands. In a well-trained model, each layer tends to concentrate on a characteristic subset of frequencies. Motivated by this insight, we analyze the training dynamics of hierarchical models and introduce two plug-and-play regularization strategies:

1. *Intra-scale frequency regularization:* a progressive spectral truncation scheme that guides each scale to specialize in its target frequency band, enabling a natural low-to-high frequency transition.

2. *Inter-scale latent regularization:* a similarity-based penalty in latent space that mitigates spectral aliasing across scales.

These regularizers operate only during training and impose no extra complexity at inference. This design effectively alleviates spectral dispersion and aliasing, leading to faster convergence and improved compression efficiency, i.e., 2.3× faster convergence and an additional 9.49% rate–distortion improvement over our baseline hierarchical model trained without regularization.

Our main contributions are threefold:

1. We conduct a spectral analysis of training dynamics in hierarchical image coding, revealing frequency dispersion, interference, and aliasing as the primary obstacles that hinder optimization efficiency and compression performance.

2. We propose two lightweight regularization strategies—spectral truncation for intra-scale specialization and similarity penalties for inter-scale coordination—that effectively mitigate these spectral issues.

3. We develop a compact hierarchical architecture which, combined with the proposed training scheme, achieves a 20.65% bitrate savings over VTM-22.0, surpassing both learned and traditional codecs and establishing a new state of the art in learned image compression.

## 2 PRELIMINARIES

### 2.1 SINGLE-SCALE IMAGE CODING

The single-scale VAE-based image codec can be generally divided into two key components. First, various nonlinear networks such as CNNs (Ballé et al., 2018; Minnen et al., 2018; Cheng et al., 2020; He et al., 2022), Transformers (Lu et al., 2021; Liu et al., 2023) or Mamba (Qin et al., 2024; Zeng et al., 2025), are employed to perform encoding and decoding transformations from input image $\mathbf{x}$ to latent $\mathbf{y}$ and from decoded $\hat{\mathbf{y}}$ to reconstruction $\hat{\mathbf{x}}$, namely $\mathbf{y} = g_a(\mathbf{x})$ and $\hat{\mathbf{x}} = g_s(\hat{\mathbf{y}})$. These networks exploit the spatial and channel-wise correlations of the image, aiming to extract the most compact latent representations. Second, deliberate probabilistic modeling processes are used to estimate the distribution of the latent variables, such as Hyperprior (Ballé et al., 2018) which further extracts abstract representations $\mathbf{z}$ from $\mathbf{y}$ to estimate the probability distribution of $\hat{\mathbf{y}}$, expressed as $\mathbf{z} = h_a(\mathbf{y})$ and $\hat{\mathbf{y}} = h_s(\hat{\mathbf{z}})$, and various context models (Cheng et al., 2020; He et al., 2022; Liu et al., 2023), which enable autoregressively modeling the distribution of latent variables in sequence.

For this single-scale structure, constrained rate-distortion optimization with a Lagrangian multiplier $\lambda$ is applied to train it in an end-to-end way, i.e.,

$$\mathcal{L}_{single} = R(\mathbf{y}) + R(\mathbf{z}) + \lambda \cdot D(\mathbf{x}, \hat{\mathbf{x}}), \quad (1)$$

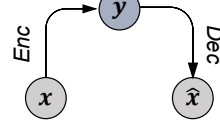

Figure 2: Diagram of single-scale codecs.

where $R$ is the bitrate consumption of encoding latent $\mathbf{y}$ or hyperprior $\mathbf{z}$ (if any), and $D$ is the distortion term between the reconstruction $\hat{\mathbf{x}}$ and the input image $\mathbf{x}$. The final goal of such single-scale codecs is to eliminate redundant information that has less impact on the loss function, ultimately extracting the most compact single-scale representation, under a preset rate-distortion trade-off. However, recent studies show that training single-scale coding frameworks is hampered by conflicting rate–distortion objectives and unstable parameter updates. These problems induce inefficient training and limited performance. Consequently, recent studies have begun to examine the training dynamics of single-scale image codecs and propose solutions such as gradient modulation (Zhang et al., 2025c), improved optimizers (Li et al., 2025a; Zhang et al., 2025b), and auxiliary training networks (Li et al., 2025b).

### 2.2 HIERARCHICAL IMAGE CODING

Hierarchical image codecs (Hu et al., 2020; 2021; Ryder et al., 2022; Duan et al., 2023b;a; Lu et al., 2024; Zhang et al., 2025a) extend single-scale models into a multi-scale framework, in which an image is represented through a hierarchy of latent variables at different resolutions. Each scale captures complementary information: higher-scale latents encode abstract, global structures, while lower-scale latents represent fine-grained, high-frequency details (Sønderby et al., 2016; Vahdat & Kautz, 2020). During compression, these latent variables are progressively predicted and entropy-coded, allowing for flexible rate allocation and efficient reconstruction. Formally, an $L$-scale hierarchy is optimized by minimizing the sum of scale-wise bitrate costs and the final reconstruction distortion, i.e.,

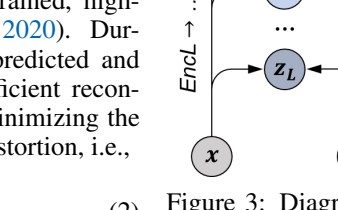

$$\mathcal{L}_{hier} = \sum_{l=1}^{L} R(\mathbf{z}_l) + \lambda \cdot D(\mathbf{x}, \hat{\mathbf{x}}). \quad (2)$$

Figure 3: Diagram of hierarchical codecs.

From a theoretical perspective, hierarchical coding naturally aligns with the frequency principle (Rahaman et al., 2019; Ronen et al., 2019; Xu & Zhou, 2021; Xu et al., 2025). Specifically, global image content, which is more efficiently captured at higher scales, predominantly lies in low-frequency

components, whereas local details at lower scales correspond to high-frequency components. This coarse-to-fine decomposition mirrors the frequency-dependent convergence dynamics observed in neural networks: deeper layers tend to model low-frequency components faster, while shallower layers capture high-frequency components gradually. Hence, hierarchical image coding can be viewed as a structural embodiment of the frequency principle, with each scale specializing in a specific spectral range, enabling efficient multiscale representation and progressive reconstruction.

However, in practice, this ideal frequency-aligned decomposition is not always perfectly realized. Latent variables at different scales may become entangled, and competing gradients during optimization can lead to overlapping spectral representations across scales. As a result, some scales may partially encode information outside their intended frequency range, and the hierarchical allocation of coarse-to-fine information may be disrupted. Understanding this deviation is crucial for analyzing hierarchical training dynamics and explaining why naive optimization sometimes fails to fully exploit the theoretical advantages of multiscale latent structures.

## 3 METHODOLOGY

### 3.1 SPECTRAL ANALYSIS OF HIERARCHICAL TRAINING DYNAMICS

To investigate hierarchical training dynamics through spectral analysis, we quantify the scale-wise contributions to the final reconstruction (see implementation details in Appendix A.2) and compute their spectral overlap with the input image. Tracking the evolution of this overlap across training epochs with a heatmap (Fig. 1) reveals a two-stage pattern:

**Early Stage:** Different scales converge to their respective frequency bands at different rates. Higher scales are more sensitive to low-frequency content and converge faster, whereas lower scales focus on high-frequency details and converge more slowly.

**Later Stage:** As training progresses, each scale stabilizes within a certain spectral range. At this point, the scales separate their spectra from one another, forming a decoupled low-to-high frequency distribution (Maaløe et al., 2019; Vahdat & Kautz, 2020).

Overall, the training process broadly follows the frequency principle. However, several localized issues still disrupt this progression. As illustrated in Fig. 1a and 1c:

**Intra-scale interference:** Spectral components become entangled with high-frequency noise and low-frequency interference. Worse still, these artifacts propagate through the hierarchy, causing the spectrum of the last scale to exhibit severe dispersion.

**Inter-scale aliasing:** Overlapping frequency bands persist across scales—for example, the second scale contains an abnormal low-frequency band that overlaps with the first, while the last scale almost entirely covers the third.

We argue that these violations of the frequency principle lead to ill-structured information hierarchies, resulting in training instability and limited performance (see Fig. 7 and 8). This observation naturally motivates the question: *Can we design explicit guidance based on the frequency principle to promote spectral convergence and decoupling, thereby mitigating interference and aliasing?*

To address this, we propose an intra-scale regularization in the early training (e.g., the first 100 epochs) to stabilize scale-wise frequency band convergence. Then we switch to an inter-scale regularization in later stages to mitigate spectral aliasing, more details are described below.

### 3.2 INTRA-SCALE FREQUENCY REGULARIZATION

To ensure that each scale converges quickly and accurately to its assigned frequency band in early training without mixing in abnormal frequency components, we design a Discrete Cosine Transform (DCT) based spectral truncation scheme. Specifically, at first, only the low-frequency components of the input are fed into the entire model, and then higher frequencies can gradually be added. This allows the topmost scale (with largest receptive field) $\mathbf{z_1}$ rapidly and fully capture low-frequency in-

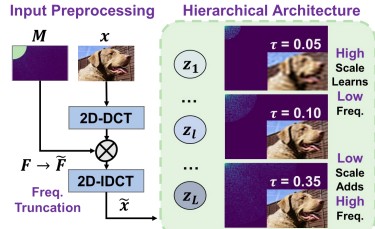

Figure 4: Pipeline of proposed intra-scale regularization.

formation, avoiding delegating low-frequency responsibility to subsequent scales. Later scales can then focus on their assigned high-frequency contents, avoiding interference of high-frequency noise across scales. The overall pipeline is illustrated in Fig. 4.

In implementation, the training data $\mathbf{x} \in \mathbb{R}^{B \times C \times H \times W}$ is first transformed into frequency domain $\mathbf{F} \in \mathbb{R}^{B \times C \times H \times W}$ through 2D-DCT as equation 3, where $B, C, H, W$ are the batch size, channel numbers, height and width, respectively.

$$\mathbf{F} = P_H \mathbf{x} P_W^\top, \quad \text{where} \begin{cases} (P_H)_{u,x} = \alpha_H(u)\cos\left(\frac{\pi(2x+1)u}{2H}\right), \\ (P_W)_{v,y} = \alpha_W(v)\cos\left(\frac{\pi(2y+1)v}{2W}\right), \end{cases} \alpha_K(k) = \begin{cases} \sqrt{\frac{1}{K}}, & k = 0, \\ \sqrt{\frac{2}{K}}, & k \geq 1, \end{cases} \tag{3}$$

where $P_H \in \mathbb{R}^{H \times H}$ and $P_W \in \mathbb{R}^{W \times W}$ are the orthonormal bases along the vertical and horizontal dimensions of DCT, respectively. $u \in [0, H-1]$ and $v \in [0, W-1]$ are the frequency indices, and the normalization term $\alpha_K(k)$ is employed to guarantee orthogonality, with $K \in \{H, W\}$.

Then, a time-varying soft radial mask $\mathbf{M}(u, v; t) \in [0, 1]$ is used for spectral truncation, defined as:

$$\mathbf{M}(u, v; t) = \max\left(0, \frac{\tau(t) - \sqrt{(\frac{u}{H})^2 + (\frac{v}{W})^2}}{\tau(t)}\right), \tag{4}$$

where the term $\sqrt{(u/H)^2 + (v/W)^2}$ represents the normalized frequency radius, and $\tau(t)$ is a scheduling function of epochs $t$ that controls the cutoff radius, it typically increases from a small initial value (e.g., $\tau(0) = 0.05$ in our scheme) to 1 in a linear manner during training. Then the original spectrum $\mathbf{F}$ can be truncated to $\widetilde{\mathbf{F}} = \mathbf{F} \cdot \mathbf{M}(u, v; t)$.

Finally, the truncated spectrum $\widetilde{\mathbf{F}}$ will be transformed back to the pixel domain via 2D-IDCT, yielding images that retain only a subset of frequency components $\widetilde{\mathbf{x}}$ for model training. More details about the implementation of 2D-DCT can be found in Appendix A.2.

$$\widetilde{\mathbf{x}} = P_H^T \cdot \widetilde{\mathbf{F}} \cdot P_W. \tag{5}$$

This low-to-high frequency learning strategy enables an incremental optimization from high to low scales: high scales fully encode the low-frequency information without leaking it to subsequent scales, while high-frequency information is gradually incorporated in lower scales on the basis of low-frequency representations. Such a schedule accounts for the varying convergence rates and sensitivities to different frequency bands across scales, as dictated by the frequency principle, thereby avoiding abnormal frequency within each scale.

### 3.3 INTER-SCALE LATENT REGULARIZATION

Then, as training processes, each scale's approximate spectral range is largely fixed. At this stage, our goal is to prevent spectral overlap between scales and to organize inter-scale information effectively. To achieve this, we introduce a regularization based on latent variables' similarity across adjacent scales, which encourages the features of neighboring scales to remain as distant as possible. This ensures that subsequent scales allocate bitrate only to frequency components not represented by preceding scales, thereby saving bitrate. The pipeline is illustrated in Fig. 5.

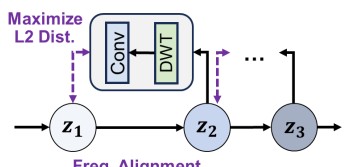

Figure 5: Pipeline of proposed inter-scale regularization.

In practice, we insert a convolutional downsampling module based on the Discrete Wavelet Transform (DWT) between consecutive latent variables only during training (disabled when inference). The lower-scale latent variable $\mathbf{z}_l$ is downsampled via DWT, which decomposes it into frequency sub-bands. We then apply a 1×1 convolution across channels to linearly map and recompose frequency sub-bands so they align with the frequency channels of higher scale latent $\mathbf{z}_{l-1}$. The feature regularization distances are measured by L2 loss and weighted into the loss for end-to-end optimization. For the topmost scale, the latent variable will be compared to the initial learnable bias prior obtained from training. In this way, the model's training loss in equation 2 is re-formulated as:

$$\mathcal{L}_{hier\_regu} = \sum_{l=1}^{L} R(\mathbf{z}_l) + \lambda \cdot D(\mathbf{x}, \hat{\mathbf{x}}) - \delta \cdot \sum_{l=1}^{L} L_2(\mathbf{z}_{l-1}, Conv_{1 \times 1}(DWT(\mathbf{z}_l))), \tag{6}$$

where $\delta$ is a weight parameter, it is fixed to 0.1 in our implementation. More details such as the implementation of DWT and different choices of $\delta$, can be found in Appendix A.2 and A.4.

By integrating such inter-scale regularization, an inter-scale penalty mechanism is established to discourage the aligned lower latent from predicting the same low-frequency content in the higher latent. Hence, the model can carefully avoid spending bitrate to encode redundant or overlapping spectral information. This guides the model to explore a more decoupled and efficient information distribution across scales, thereby saving overall bitrate cost.

## 4 EXPERIMENTS

### 4.1 SETUP

**Base Model Design:** To thoroughly investigate the training dynamics of hierarchical coding, we design a lightweight 4-scale hierarchical image codec, DHIC (Deep Hierarchical Image Coding). Unlike prior approaches (Duan et al., 2023b;a; Lu et al., 2024; Zhang et al., 2025a), our base model, a.k.a. DHIC-Base, adopts only a single latent block per scale and replaces heavy backbones (e.g., Transformer or Mamba (Gu & Dao, 2023)) with simple CNNs. This design eliminates performance confounds introduced by complex architectures, allowing us to focus purely on the effect of training with and without regularization as mentioned above. We refer to the DHIC model trained with the proposed regularization schemes as the DHIC-Regu. Network details are provided in Appendix A.2.

**Training Settings:** We train our models, e.g., DHIC-Base, and DHIC-Regu, on the mixed dataset introduced in Jiang et al. (2025), which comprises images selected from Flickr20K (Lim et al., 2017), DIV2K (Agustsson & Timofte, 2017), COCO2017 (Lin et al., 2014), and ImageNet (Deng et al., 2009). During pretraining, images are randomly cropped into $256 \times 256$ patches with a batch size of 32, while finetuning is performed on $512 \times 512$ crops with a batch size of 4. The training procedure is conducted on a single NVIDIA RTX 4090 GPU using `PyTorch` and the Adam optimizer. The learning rate is initialized at $1e{-}4$ and gradually reduced to $1e{-}5$ via a `ReduceLROnPlateau` scheduler during pretraining, and further decreased to $1e{-}6$ in the finetuning stage. In addition, our codec supports variable bitrate, with the Lagrangian factor $\lambda$ ranging from 64 to 4096.

**Test Settings:** We conduct tests on three widely used test datasets: the Kodak dataset (Eastman Kodak Company, 1993), which consists of 24 images with a resolution of $512 \times 768$ (or $768 \times 512$); the CLIC professional Valid dataset (Toderici et al., 2020), which contains 41 high-quality images of varying resolutions; and the Tecnick dataset (Asuni et al., 2014), which contains 100 images at a resolution of $1200 \times 1200$. For rate–distortion evaluation, we report bitrate in BPP and distortion using either PSNR or MS-SSIM (Wang et al., 2004). In addition, we adopt the Bjøntegaard Delta Rate (BD-Rate) (Bjontegaard, 2001) to measure gains to the anchor codec. Tests using 4K or 1080p images are provided in the Appendix A.4.

We also report the model's parameter size (M), computational complexity in terms of KMACs/pixel (kilo multiply–accumulate operations per pixel), and the encoding/decoding time (ms) to assess the complexity of the codecs. All evaluations are performed on a platform equipped with a single NVIDIA RTX 3090 GPU and an Intel Xeon Gold 6430 CPU.

### 4.2 EVALUATION RESULTS

**Compression Performance:** We comprehensively evaluate the proposed DHIC-Regu, DHIC-Base, a series of state-of-the-art single-scale learned codecs (He et al., 2022; Liu et al., 2023; Li et al., 2023; Qin et al., 2024; Fu et al., 2024; Li et al., 2025c; Zeng et al., 2025; Jiang et al., 2025), and the representative hierarchical image codec QARV (Duan et al., 2023a). As summarized in Table 1, and using VTM-22.0 as the anchor baseline, DHIC-Regu achieves the best BD-Rate performance across all three datasets, i.e., –19.73%, –18.13%, and –24.09%. Notably, our method demonstrates even greater advantages on high-resolution images compared to the latest single-scale codec HPCM-Large (Li et al., 2025c), as detailed in Appendix A.4. Moreover, compared to DHIC-Base without regularization, DHIC-Regu achieves an additional 9.49% bitrate reduction over VTM-22.0, without introducing any extra testing complexity. This clearly demonstrates the effectiveness of our regularization design. We also validate its effectiveness on QARV, as reported in Appendix A.4.

Table 1: **Compression performance and complexity comparison of learned image codecs across multiple datasets** (Anchor: VTM-22.0).

| Model | Enc. Time (ms) | Dec. Time (ms) | KMACs (/pixel) | Params (M) | Kodak (%) | CLIC Pro (%) | Tecnick (%) | Avg. (%) |
|---|---|---|---|---|---|---|---|---|
| ELIC (CVPR'22) | 43.78 | 48.15 | 573.88 | 36.93 | -3.22 | -3.89 | -4.57 | -3.89 |
| TCM-Large (CVPR'23) | 153.57 | 142.83 | 1823.58 | 76.57 | -9.97 | -9.65 | -13.24 | -10.95 |
| MLIC++ (NCW ICML'23) | 191.46 | 186.08 | 1282.81 | 116.72 | -11.83 | -12.18 | -17.25 | -13.75 |
| FLIC (ICLR'24) | > 1000 | > 1000 | 1096.04 | 70.97 | -12.97 | -10.53 | -15.82 | -13.11 |
| WeConvene (ECCV'24) | 333.45 | 227.41 | 2343.13 | 107.15 | -6.98 | -8.54 | -10.81 | -8.78 |
| MambaVC (Arxiv'24) | 137.88 | 124.95 | 813.80 | 53.32 | -8.72 | -5.66 | -8.63 | -7.67 |
| MambaIC (CVPR'25) | 156.82 | 113.07 | 1284.86 | 75.78 | -15.12 | -9.98 | -13.65 | -12.92 |
| HPCM-Large (ICCV'25) | 117.40 | 112.27 | 1261.29 | 89.71 | -19.19 | -18.37 | -22.20 | -19.92 |
| QARV (TPAMI'24) | 158.42 | 71.61 | 718.96 | 93.4 | -5.81 | -6.91 | -8.88 | -7.20 |
| DHIC-Base (Ours) | 102.46 | 68.48 | 977.73 | 106.93 | -9.62 | -10.79 | -13.06 | -11.16 |
| DHIC-Regu (Ours) | 102.46 | 68.48 | 977.73 | 106.93 | -19.73 | -18.13 | -24.09 | -20.65 |

**Complexity:** We first examine the training complexity. With the proposed regularization, DHIC-Regu converges within approximately 3.8 days, which is highly competitive compared to both single-scale codecs (e.g., TCM (Liu et al., 2023)) and hierarchical codecs (e.g., QARV), each of which typically requires nearly 10 days to complete training.

For testing complexity, we compute the time overhead, computational cost, and parameter counts for each codec, as shown in Table 1. For fairness, these metrics are re-evaluated on the same platform with open-source implementations whenever available, while reported data from original papers are adopted otherwise. As observed, our hierarchical codec achieves lower complexity than the recent performance-leading methods, such as MLIC++ (Jiang et al., 2025) and HPCM-Large (Li et al., 2025c), while also offering superior performance. This benefits from both our lightweight network design and training-only regularizers. Furthermore, the hierarchical codec naturally provides a more efficient context in both spatial and frequency domains, thereby reducing the overhead of complex autoregressive contexts and achieving the fastest decoding speed at similar complexity.

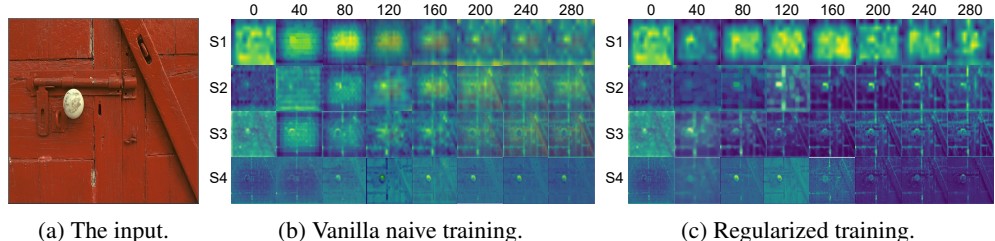

(a) The input.  (b) Vanilla naive training.  (c) Regularized training.

Figure 6: **Visualization of scale-wise latent variables during two different training processes.** The source image is *Kodak002.png* (Eastman Kodak Company, 1993). The horizontal axis denotes the training epochs, while the vertical axis corresponds to the hierarchical scales.

**Qualitative Results:** To intuitively validate the effectiveness of our proposed regularization, we visualize the hierarchical information structure under naive and regularized training via scale-wise latent variables (Fig. 6). The results show that: (a) Under naive training, latent variables across scales remain entangled and indistinct. Higher scales (e.g., S1) fail to capture global semantics, while lower scales (e.g., S4) cannot effectively represent high-frequency structures, instead showing scattered, noisy patterns. The model also exhibits grid-like artifacts and central blur, and structured features emerge only after the 120th epoch, without achieving clear scale decoupling. (b) In contrast, with regularized training, distinct latent variables emerge as early as the 40th epoch. These representations are progressively refined and naturally decoupled, resulting in a clear coarse-to-fine information hierarchy.

Overall, the visualizations confirm that the proposed regularization mitigates spectral issues such as dispersion, noise, and aliasing, thereby promoting the emergence of well-structured multiscale representations and fully unlocking the capacity of hierarchical coding architectures.

## 4.3 Deep Dive

**Intra-Scale Training Dynamics:** According to the frequency principle, models tend to learn low-frequency information first during training, as it reduces loss more effectively at lower bitrate cost. However, in the absence of explicit low-frequency constraints, each scale attempts to capture information across the full frequency spectrum in a naive manner. This mixing of frequencies prevents scales from specializing in their most relevant bands. For example, higher scales often attempt to encode high-frequency details that they are ill-suited for, leading to misrepresentation that propagates backward and ultimately

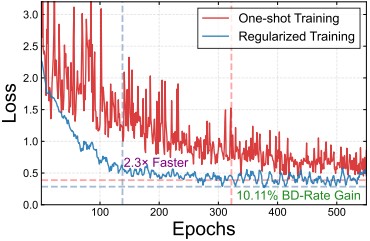

Figure 7: Loss vs. training epochs.

degrades both training stability and reconstruction quality. To illustrate this effect, we visualize the training loss curve in Fig. 7, which shows large fluctuations without substantial loss reduction during the first 100 epochs. By contrast, when explicit low-to-high frequency guidance is introduced, each scale quickly concentrates on its assigned frequency range, resulting in faster convergence (Fig. 1b) and effectively mitigating the instability and inefficiency in naive training.

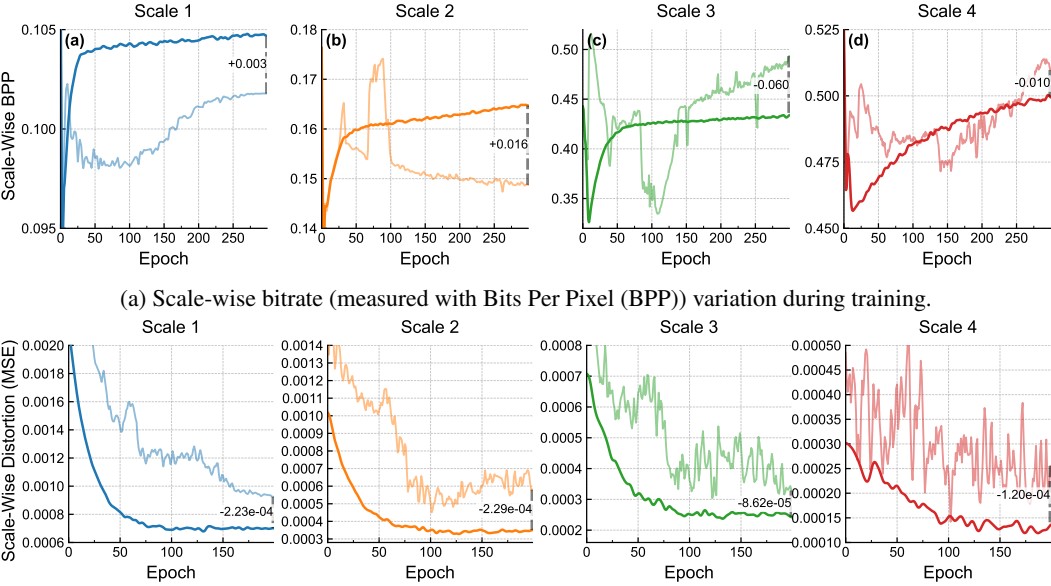

(a) Scale-wise bitrate (measured with Bits Per Pixel (BPP)) variation during training.

(b) Scale-wise distortion (measured with Mean Squared Error (MSE)) variation during training.

Figure 8: **Scale-wise rate-distortion change during training.** The light-colored curves represent the naive training, while the dark-colored curves show the regularized one. Differences in layer-wise bitrate and distortion are marked with dashed lines and text annotations in each subplot.

**Inter-Scale Training Dynamics:** Due to the inherent dependencies across scales in hierarchical architectures, latent variables at higher scales are conditioned on those from earlier ones, leading to mutual interference in scale-wise rate–distortion optimization. As a result, naive optimization following equation 2 struggles to establish an efficient hierarchical information structure. From a spectral perspective, this manifests as *spectral aliasing* (Fig. 1), where multiple scales redundantly encode overlapping frequency components. Such redundancy wastes additional bitrate and ultimately limits the achievable rate–distortion performance.

To further examine the effect on training dynamics, we visualize the scale-wise rate and distortion curves in Fig. 8. The results reveal that, during training, the model persistently reallocates bitrate across scales, preventing the scale-wise bitrate from steadily increasing or stabilizing. Instead, it undergoes frequent fluctuations and abrupt shifts. In parallel, scale-wise distortion also exhibits pronounced oscillations. These instabilities intensify at deeper scales, ultimately causing excessive bitrate consumption and higher distortion. Such behavior validates that inter-scale spectral aliasing severely disrupts training and degrades performance. In contrast, when inter-scale regularization is

introduced, this aliasing is largely mitigated (see Fig. 1b), resulting in markedly improved training stability and superior rate–distortion performance.

## 4.4 ABLATION STUDIES

**The separate effects of two regularization methods:** Although both of the proposed regularization methods aim to address the issues observed in spectral analysis, their effects differ. Table 2 presents the individual impact of each method on accelerating training and improving performance. It can be observed that applying intra-scale regularization solely in the early stages of training primarily accelerates model training but does not significantly improve the rate-distortion performance after full training. On the other hand, applying inter-scale regularization solely in the subsequent stages will somewhat slow down convergence but provide a substantial performance boost. Furthermore, both of these regularizers are essentially designed to guide hierarchical optimization to better follow the frequency principles, so they can complement each other to some extent. When both regularization methods are combined, the model achieves a synergistic effect, resulting in faster convergence and improved final performance.

Table 2: Ablation study of the separate effects of two regularizers (Baseline: the naive optimized model).

| Regularization | Acceleration | BD-Rate (%) |
|---|---|---|
| Intra-Scale | 1.84× | -1.07 |
| Inter-Scale | 0.91× | -7.66 |
| Both | 2.30× | -10.11 |

In addition, we also examine the separate effects of the two regularizers on QARV (Duan et al., 2023a) and obtain similar conclusions. The detailed results are provided in Appendix A.4.

**The effect of different regularization implementation settings:** We conduct ablation studies to compare the effects of different implementations of two regularization methods. Specifically:

1. **The scheduler during the early-stage DCT truncation:** In our implementation, the high-frequency components are linearly increased from 0.05 to 1.0 during the first 100 epochs. We further explore alternative scheduling strategies, including different initialization values and exponential growth modes, as reported in Table 3a. Empirically, the proposed linear schedule yields the best trade-off between training efficiency and final performance.

2. **Inter-scale latent regularization during subsequent training:** In our approach, latent variables from lower scales are downsampled using a convolutional layer combined with wavelet transformation, followed by L2 alignment with the preceding scale. We also investigate alternative strategies, including standard strided convolution, downsampling followed by convolution, and feature alignment using L1 loss or cosine similarity, as summarized in Table 3b. Empirically, the proposed scheme achieves the best performance. Theoretical analysis of such superiority can be found in Appendix A.4.

Table 3: **Ablation studies of intra-scale and inter-scale regularization implementations** (Baseline: the naive trained model, best implementation approaches are marked in blue color).

(a) Intra-scale regularization (First 100 epochs).

| Implementation | Acceleration | BD-Rate (%) |
|---|---|---|
| 0.025→1.0 (linear) | 1.62× | -1.01 |
| 0.05→1.0 (linear) | 1.84× | -1.07 |
| 0.1→1.0 (linear) | 1.77× | -1.05 |
| 0.025→1.0 (exp) | 1.49× | -0.93 |
| 0.05→1.0 (exp) | 1.60× | -0.98 |
| 0.1→1.0 (exp) | 1.53× | -1.02 |

(b) Inter-scale regularization (Remaining epochs).

| Implementation | Acceleration | BD-Rate (%) |
|---|---|---|
| Conv w/ Stride | 0.87× | -5.49 |
| Down + Conv | 0.90× | -5.22 |
| DWT + Conv | 0.91× | -7.66 |
| L1 | 0.82× | -7.07 |
| L2 | 0.91× | -7.66 |
| Cos Similarity | 0.85× | -6.55 |

More ablation studies on regularization setups and modules design, are detailed in Appendix A.4.

## 5 CONCLUSION

In this work, we tackle the fundamental optimization challenges inherent in hierarchical image compression by diagnosing and countering spectral dispersion and aliasing phenomena that arise during

training. Our spectral analysis reveals that these issues lead to inefficient training and degraded rate-distortion performance. To address these issues, we introduce two complementary regularization strategies: a low-to-high frequency-aware truncation mechanism that guides each intra-scale toward its intended spectral band, and an inter-scale similarity constraint that prevents the encoding of redundant frequency information across scales. These training-only techniques yield a model that converges $2.3\times$ faster and achieves an additional 9.49% improvement in rate-distortion performance over VTM-22.0, culminating in an average 20.65% bitrate savings over VTM-22.0—setting a new state-of-the-art in learned image compression. Our approach not only demonstrates the untapped potential of hierarchical architectures but also provides a principled spectral perspective for optimizing multi-scale latent models, opening avenues for future research in efficient neural compression.

## ETHICS STATEMENT

This work focuses on hierarchical image compression with the goal of improving compression efficiency from a spectral regularization perspective. All experiments are performed using publicly available datasets—including MLIC-Train-100k, UVG, CLIC professional Valid, Tecnick, and LIU4K-v2—which consist of non-sensitive and openly accessible data. We acknowledge that compression algorithms inevitably introduce distortions, which could affect the reliability of image interpretation in safety-critical applications such as medical imaging or autonomous systems. We therefore emphasize the importance of responsible deployment of the proposed method, particularly in high-stakes domains where inaccuracies may carry significant consequences.

## REPRODUCIBILITY STATEMENT

To support reproducibility, we include detailed descriptions of the model architectures, training configurations, and evaluation protocols. All experiments were conducted under controlled settings with fixed random seeds and hardware platform. Furthermore, we are committed to releasing the complete source code publicly upon acceptance of the paper to facilitate replication and encourage further research in this direction.

## ACKNOWLEDGEMENT

This work was supported in part by Natural Science Foundation of China (Grant No. 62431011, 62401251) and Natural Science Foundation of Jiangsu Province (Grant No. BK20243038). The authors would like to express their sincere gratitude to the Interdisciplinary Research Center for Future Intelligent Chips (Chip-X) and Yachen Foundation for their invaluable support.

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

# A APPENDIX

## A.1 PRELIMINARY OF FREQUENCY PRINCIPLE

In this section, we supplement more prior works on spectrum bias, frequency principle and hierarchical architecture related to spectrum analysis and training dynamics to further help understand the theoretical basis of our work.

Frequency principle, also referred to as the spectral bias theory, denotes the empirically observed tendency of deep neural networks to learn low-frequency components of a target function earlier and faster than high-frequency components. This phenomenon was first systematically identified by (Rahaman et al., 2019), which demonstrated—via Fourier-domain analyses on synthetic sinusoidal targets and examinations of manifold geometry—that low-frequency components not only converge more quickly but are also more robust to parameter perturbations, thereby providing an empirical account of networks' implicit smoothing behavior. Subsequent works formalized and generalized this observation, including Xu et al. (2019); Xu & Zhou (2021); Xu et al. (2025) proposed that the same low-to-high frequency learning order appears across multiple architectures and standard vision benchmarks, arguing that the smoothness properties of common activations help explain the bias and its implications for generalization. From a theoretical perspective, the Neural Tangent Kernel (NTK) regime (Cao et al., 2021) proves that, considering the infinite-width linearization of training, training dynamics admit a spectral decomposition in which the kernel's eigenvalues determine convergence rates: directions associated with large eigenvalues (often low-frequency components) are learned rapidly, whereas directions associated with small eigenvalues (often high frequencies) are learned slowly.

Motivated by these insights, subsequent works began to regular the training process of deep neural network. For instance, Tancik et al. (2020) demonstrated that mapping low-dimensional inputs into a higher-dimensional sinusoidal feature space yields an effective kernel with controllable bandwidth, substantially improving MLPs' ability to fit high-frequency functions and underpinning the success of positional encodings in implicit-representation systems (Xie et al., 2023; Liu et al., 2024; Shi et al., 2024; 2025).

Our spectrum analysis and regularization methods are also based on the above strong conclusion—*Different scales converge to different frequency bands at different rates*, which has been widely proven in prior works, specifically including:

1. Studies on neural network spectral bias or spectral principles Rahaman et al. (2019); Xu et al. (2019); Xu & Zhou (2021); Xu et al. (2025) have shown that networks exhibit a preference for low-frequency functions during training. Low frequencies are learned earlier and faster than high frequencies in deeper module of the network. Mapping this finding to our network structure can explain why different scales exhibit different convergence speeds for different frequency bands.

2. Existing HVAE Sønderby et al. (2016); Vahdat & Kautz (2020) and hierarchical coding works Duan et al. (2023b;a) both indicate that different hierarchical scales tend to capture information at different frequencies: low-resolution scales capture more abstract global information (typically low-frequency structures), whereas high-resolution scales handle local details and residual information (typically high-frequency).

Furthermore, our existing experiments can also support the conclusion. For instance, Figure 1 shows that scale 1 has essentially converged to its corresponding spectral position (and no longer increases) at around epoch 30; subsequently, scale 2 converges at around epoch 50, scale 3 at around epoch 80, and scale 4 at around epoch 100. The original training loss curves in Fig. 8 also roughly follow this trend, which validates the above claim.

Taken together, these theoretical analysis and visualizations demonstrate, in our hierarchical structure, different scales will converge to different frequency bands at different rates: deeper, lower-resolution scales converge faster to low-frequency regions, and shallower, higher-resolution scales then gradually converge to their corresponding high-frequency positions on that basis.

## A.2 More Implementation Details

In this section, we provide additional implementation details of our methods and experiments, including network design, benchmark implementation, and testing commands. The details are presented as follows.

**Network Design:** Fig. 9 shows the Architecture of our hierarchical image codec used in this work. It consists of four scales ($L = 4$), specifically including a bottom-up encoding pathway and a top-down entropy model, plus a decoding pathway. Given an input $\mathbf{x}$, we first apply `Patchify` operations to reorganize the feature channels and spatiality. Then the reorganized feature will be fed into four cascaded encoder stages that downsample and extract features gradually, producing encoded features $\mathbf{r_{1:L}}$ at 1/8, 1/16, 1/32, and 1/64 resolution levels of input $\mathbf{x}$. Each encoder stage is formed by two branches: a main cascaded convolutional branch and a single-scale, wavelet-based convolutional branch. The two branches are summed to better capture multi-scale and multi-frequency components. At the end of the encoding pathway, a convolutional block will produce two 1/64-resolution bias features, $Em\ bias$ and $Dec\ bias$. These features seed the subsequent entropy models and decoders and are directly quantized and entropy-coded into the bitstream.

Starting from the $Em\ bias$ and $Dec\ bias$, each level is processed by its entropy model branch in sequence first. The entropy model conditions on the entropy-decoded features from the previous level $\mathbf{e}_{l-1}$ (for the first scale, it is $Em\ bias$) as prior $\mathbf{p}_l$ to estimate the posterior distribution $\mathbf{q_l}$ of the current level's encoded features $\mathbf{r_l}$. Then we compute the KL divergence between the posterior and the prior $D_{KL}(\mathbf{q}_l \parallel \mathbf{p}_l)$, yielding the scale-wise rate term $R_l$. We then sample the latent $\mathbf{z}_l$ from posterior $\mathbf{q}_l$. Adding the sampled $\mathbf{z}_l$ to the previous level's entropy-decoded features $\mathbf{e}_{l-1}$ produces the current level's entropy-decoded features $\mathbf{e}_l$. Next, the decoder at each level concatenates the previous level's decoded features $\mathbf{d}_{l-1}$ (for level one this is $Dec\ bias$) with the current level's entropy-decoded features $\mathbf{e}_l$ and fuses them to produce the level's decoding features $\mathbf{d_l}$. At the end of each level, the entropy-decoded features $\mathbf{e}_l$ and decoding features $\mathbf{d}_l$ are upsampled and refined, then passed to the next level. After four levels, the final decoding features $\mathbf{d}_L$ are `Un-Patchified` to produce the final reconstructed image.

To focus on the training dynamics of the hierarchical structure, we adopt a simple CNN backbone and design the `Basicblock` shown in Fig. 9. Each `Basicblock` consists of a re-parameterized convolution, a `SiLU` activation, and a Feed-Forward Neural Network (FFN) module in cascade, with a residual shortcut. The block is lightweight and compact, which preserves basic feature-expressive ability while keeping inference efficient.

**Progressive Decoding:** The hierarchical structure naturally supports a progressive decoding function, which enables independent decoding with scale-wise latent. We can leverage this characteristic to quantify scale-wise contribution to the final reconstruction. Specifically, for scale $l$, we decode normally for previous scales $\leq l$. For subsequent scales $> l$, we replace their posteriors $\mathbf{q}_{>l}$ with the mean of priors $\mathbf{p}_{>l}$, so no information from the bitstream of those scales is leveraged. Then a scale-wise progressively decoded reconstruction $\mathbf{x_l^P}$ can be obtained. In this case, we can define the residual $\mathbf{x_l^P} - \mathbf{x_{l-1}^P}$ as the contribution of the latent variable at the $l$-th scale, denoted as $I_l^p$. Next, we transform $I_l^p$ and the input image $\mathbf{x}$ to the frequency domain with the discrete cosine transform (DCT). We then compute their spectral overlap and plot it as a heatmap, as illustrated in Fig. 1.

**Implementation Details of DCT and DWT:** For the intra-scale regularization, we adopt the standard orthogonal DCT-II basis with a full-image size. After computing the DCT coefficients, a frequency mask is applied that preserves only a subset of low-frequency components while zeroing out the others. The mask is progressively expanded during training: in the linear schedule, the retained frequency ratio grows linearly from 0.05 to 1.0 over the first 100 epochs; in the exponential schedule, the ratio follows $r(t) = r_0 + (1.0 - r_0) \cdot \frac{1-\exp(-0.02 \times t)}{1-\exp(-0.02 \times 100)}$ such that $r(t)$ initializes to $r_0$ (i.e., 0.05) and approaches 1.0 at epoch 100.

For the inter-scale regularization, we employ a Haar wavelet downsampling block (only retain and concatenate LL, LH, and HL components), followed by an $1 \times 1$ convolution to align feature resolutions before computing similarity. We also experimented with alternative implementations (e.g., stride-2 convolution, downsample followed by convolution), and found that the combination of Haar wavelet and $1 \times 1$ convolution provided the best performance gain.

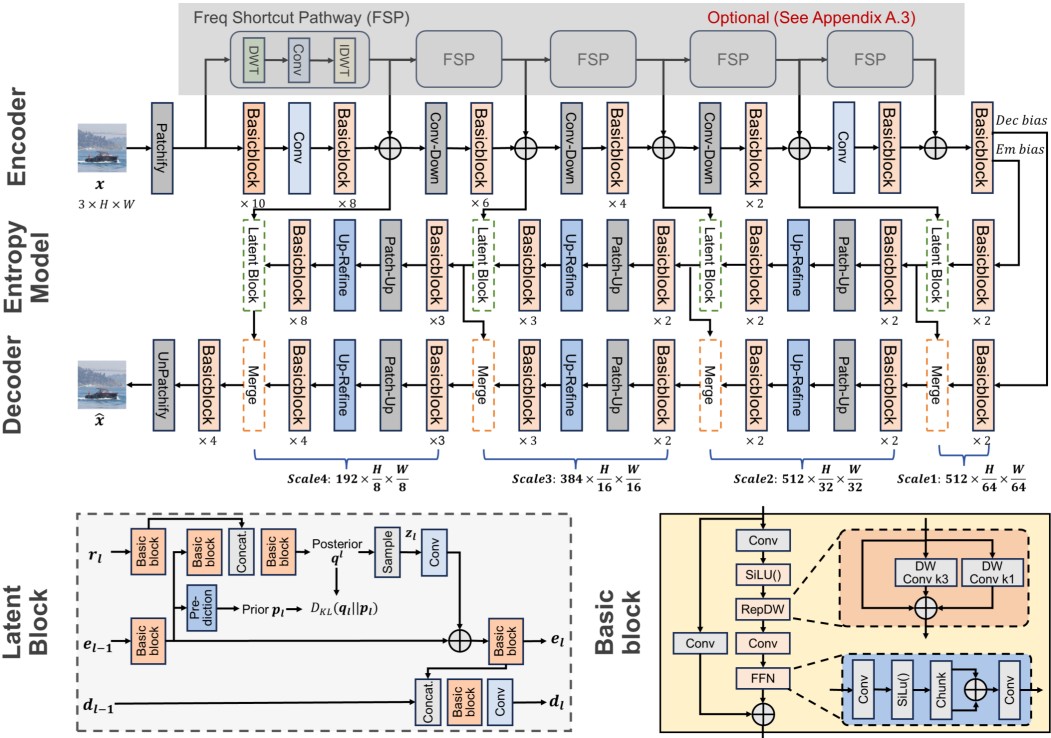

Figure 9: **Our proposed lightweight hierarchical image codec architecture.** The above is the overall network framework, where the three rows from top to bottom are the encoding pathway, entropy model pathway, and decoding pathway. And the shaded area represents the FSP module, which has been proven to be unnecessary in Appendix A.4. The lower left corner shows the network structure of the latent block in the entropy model, while the bottom right corner shows the structure of the basic model employed in our whole architecture.

**Further explanation of the inter-scale regularization:** We perform an explicit frequency-aware alignment by applying a DWT-based transform to the lower latent, which decomposes it into frequency sub-bands. We then apply a 1×1 convolution across channels to linearly map and recompose frequency sub-bands so they align with the higher-scale latent's frequency channels. The inter-scale penalty then discourages the aligned coarser latent from predicting the same low-frequency content in the higher latent. Thus, the higher-scale latent retains the difficult-to-predict high-frequency components while redundant low-frequency parts are suppressed, alleviating spectral aliasing.

**Implementation Details of Fig. 1:** Figure 1 is a heatmap that measures the degree of spectral overlap between the scale-wise latent and the original image during training (in practice, this is reflected through the progressive decoded reconstruction of each scale). The horizontal axis represents the number of training epochs, and the vertical axis represents the normalized frequency range. A point at (epoch, frequency position) indicates the spectral-overlap intensity between the latent of a certain scale and the original image at that frequency when training reaches the current epoch. The specific implementation steps are as follows:

1. Using progressive decoding, obtain $\hat{x}_l$, which is normally decoded up to layer $l$, while the remaining $L - l$ layers directly use the mean value of prior;

2. Define $I_l = \hat{x}_l - \hat{x}_{l-1}(I_0 = \hat{x}_0)$ as the mutual information between the reconstruction using only the latent of scale $l$ and the original image;

3. Apply a 2D-DCT transform to the original image $x$ and each layer's mutual information $I_l$ to obtain their spectra $F_x$ and $F_l$;

4. Perform radial binning on the 2D spectrum and convert it into a 1D form;

5. Normalize the frequency range and compute the spectral-overlap degree between the mutual information $I_l$ and the original image $x$ in each bin: $E = \sum_{b_i}^{b_{i+1}} \frac{F_l^{b_i}}{F_x^{b_i}}$;

6. Finally, use the matplotlib plotting script to generate the heatmap for each scale.

Besides, We also add the `matplotlib` plotting script for Figure 1 as follow.

```python
import numpy as np
import matplotlib.pyplot as plt
from numpy.linalg import norm

data = np.load('spectra_epochxxx.npz')
P = data['P']
freq_axis = data['freq']

epochs, layers, bins = P.shape

fraction = P / (P.sum(axis=1, keepdims=True) + 1e-12)

centroids = np.zeros((epochs, layers))
leakage = np.zeros((epochs, layers))
r_cut = 0.3
for e in range(epochs):
    for k in range(layers):
        Pk = P[e, k, :]
        centroids[e, k] = (freq_axis * Pk).sum() / (Pk.sum() + 1e-12)
        leakage[e, k] = Pk[freq_axis > r_cut].sum() / (Pk.sum() + 1e-12)

similarity = np.zeros((epochs, layers, layers))
for e in range(epochs):
    for i in range(layers):
        for j in range(layers):
            vi = P[e, i, :]
            vj = P[e, j, :]
            denom = (norm(vi) * norm(vj) + 1e-12)
            similarity[e, i, j] = np.dot(vi, vj) / denom

fig, axes = plt.subplots(1, min(4, layers), figsize=(12, 4), sharey=True)
for k in range(min(4, layers)):
    data = fraction[:, k, :].T
    im = axes[k].imshow(data, aspect='auto', origin='lower', vmin=data.
    ↪ min(), vmax=data.max(),
                        extent=[0, epochs - 1, freq_axis[0], freq_axis
    ↪ [-1]])
    axes[k].set_xlabel('Epoch', fontsize=10)
    if k == 0:
        axes[k].set_ylabel('Normalized Radial Freq.', fontsize=10)
    axes[k].set_title(f'Scale {k+1}', fontsize=10)
    axes[k].tick_params(axis='both', labelsize=8)

cbar_ax = fig.add_axes([0.92, 0.15, 0.02, 0.7])
fig.colorbar(im, cax=cbar_ax)
plt.tight_layout(rect=[0, 0, 0.9, 1])
plt.savefig('./fig2a.pdf', bbox_inches='tight', transparent=False,
    ↪ pad_inches=0.04)
plt.close(fig)
```

**Benchmarks:** For the traditional image codecs, VTM-22.0, We directly employ its open-source standard testing software https://vcgit.hhi.fraunhofer.de/jvet/VVCSoftward_VTM, and the testing commands are illustrated as follows:

```
# Convert RGB image to YUV444 format
ffmpeg -i [input_file] \
```

```
-s [width]x[height] \
-pix_fmt yuv444p \
[output_file]
```

```
# Encode
VVCSoftware_VTM/bin/EncodecAppStatic -i [input_file] \
-c [config_file] \
-q [quality] \
-o [output_path] \
-b [bitstream_file] \
-wdt [image_width] \
-hpt [image_height] \
-fr 1 \
-f 1 \
--InputChromaFormat=444 \
--InputBitDepth=8 \
--ConformanceWindowMode=1 \
```

```
# Decode
VVCSoftware_VTM/bin/DecodecAppStatic -b [bitstream_file] \
-o [output_file] \
-d 8
```

For the other learned image codecs, whenever possible, we use the open-source code to conduct evaluation under the same setups.

## A.3 MORE RESULTS

In this section, we present additional comparisons on both objective and subjective quality, including rate–distortion curves optimized with MSE and MS-SSIM, respectively, as well as visual comparisons of reconstructions from different codecs. We also provide progressive decoding results at each scale before and after applying the proposed regularization. The results are as follows.

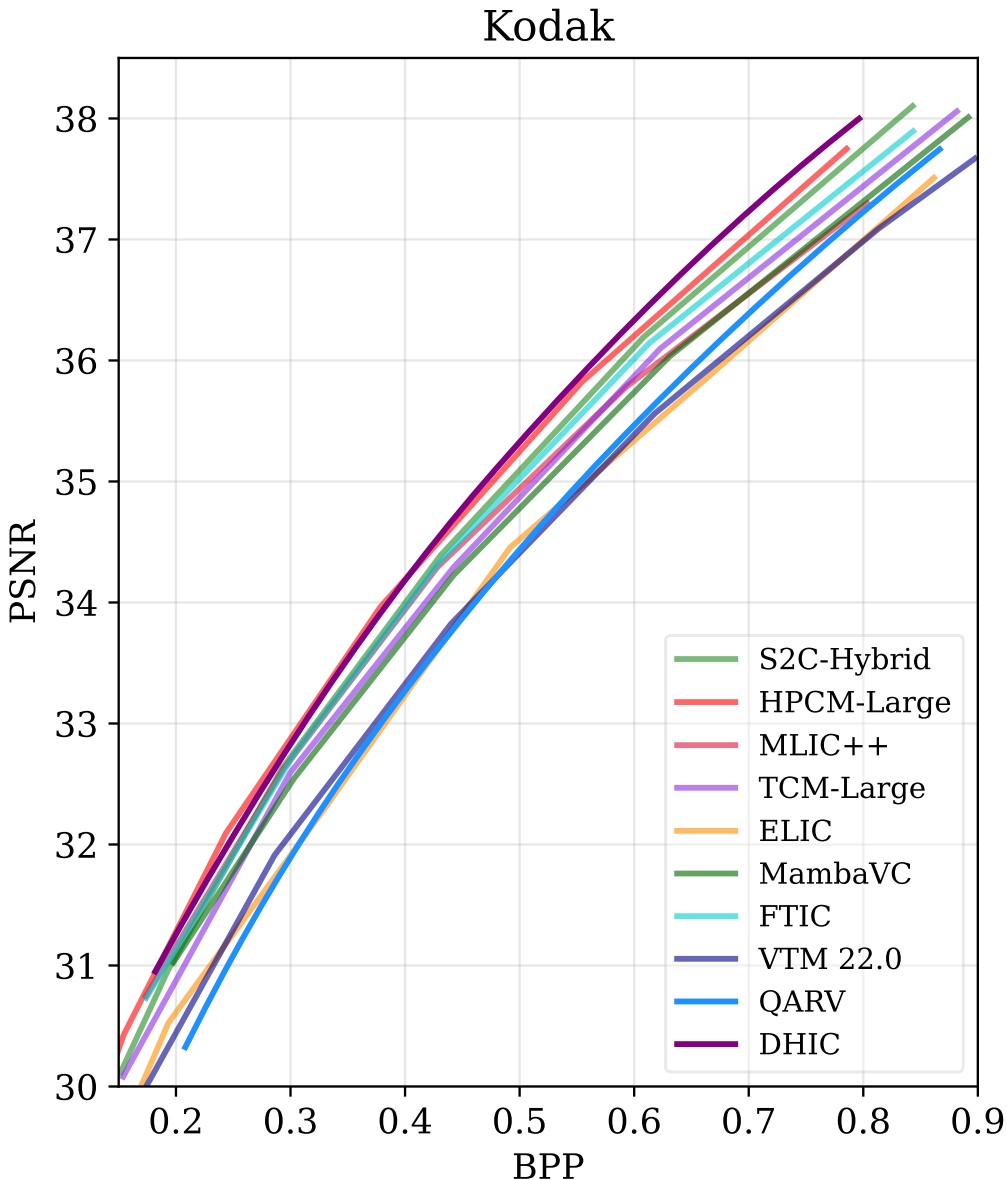

Figure 10: Rate-Distortion curves on Kodak dataset, all models are optimized with MSE.

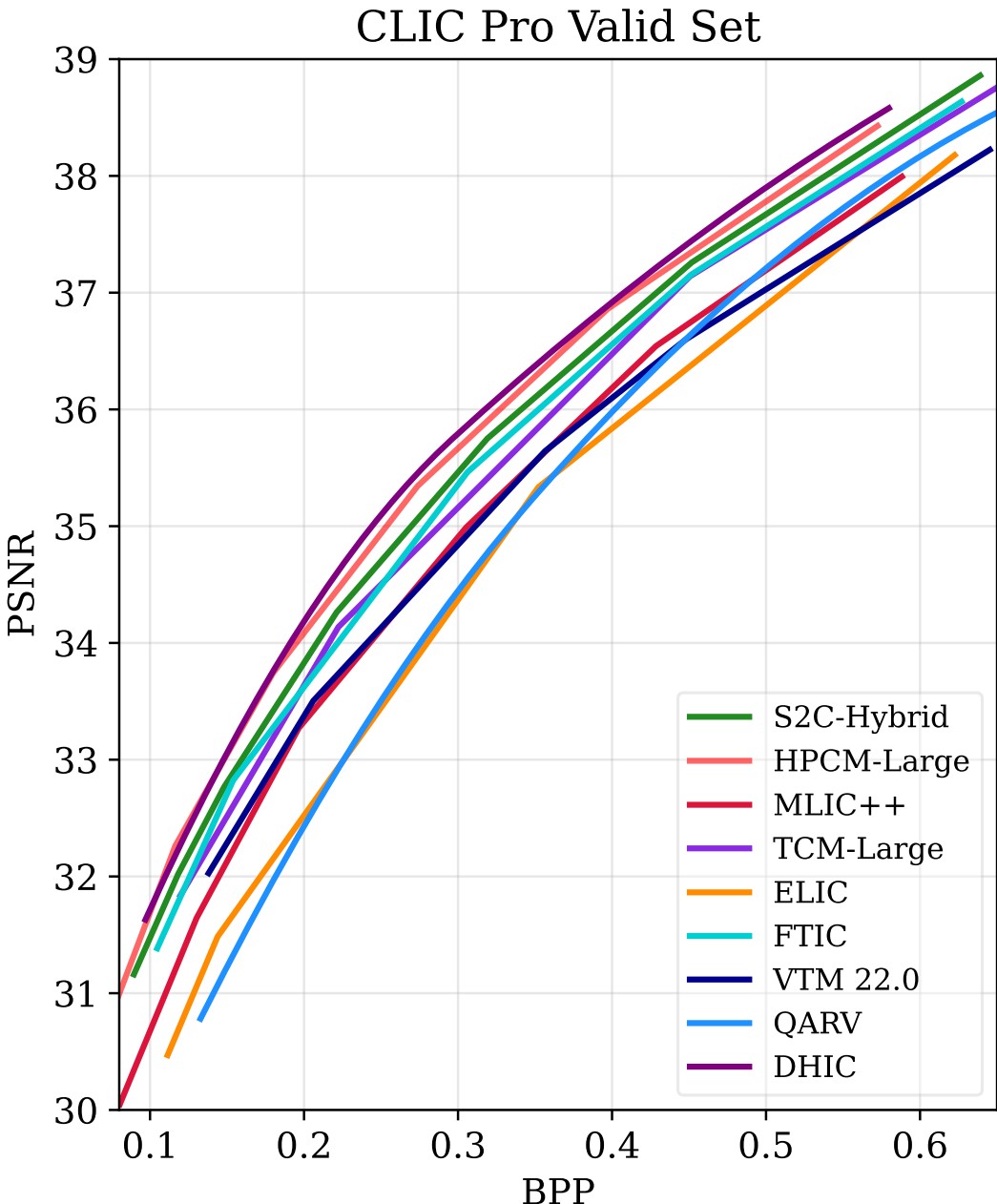

Figure 11: Rate-Distortion curves on CLIC Professional Valid dataset, all models are optimized with MSE.

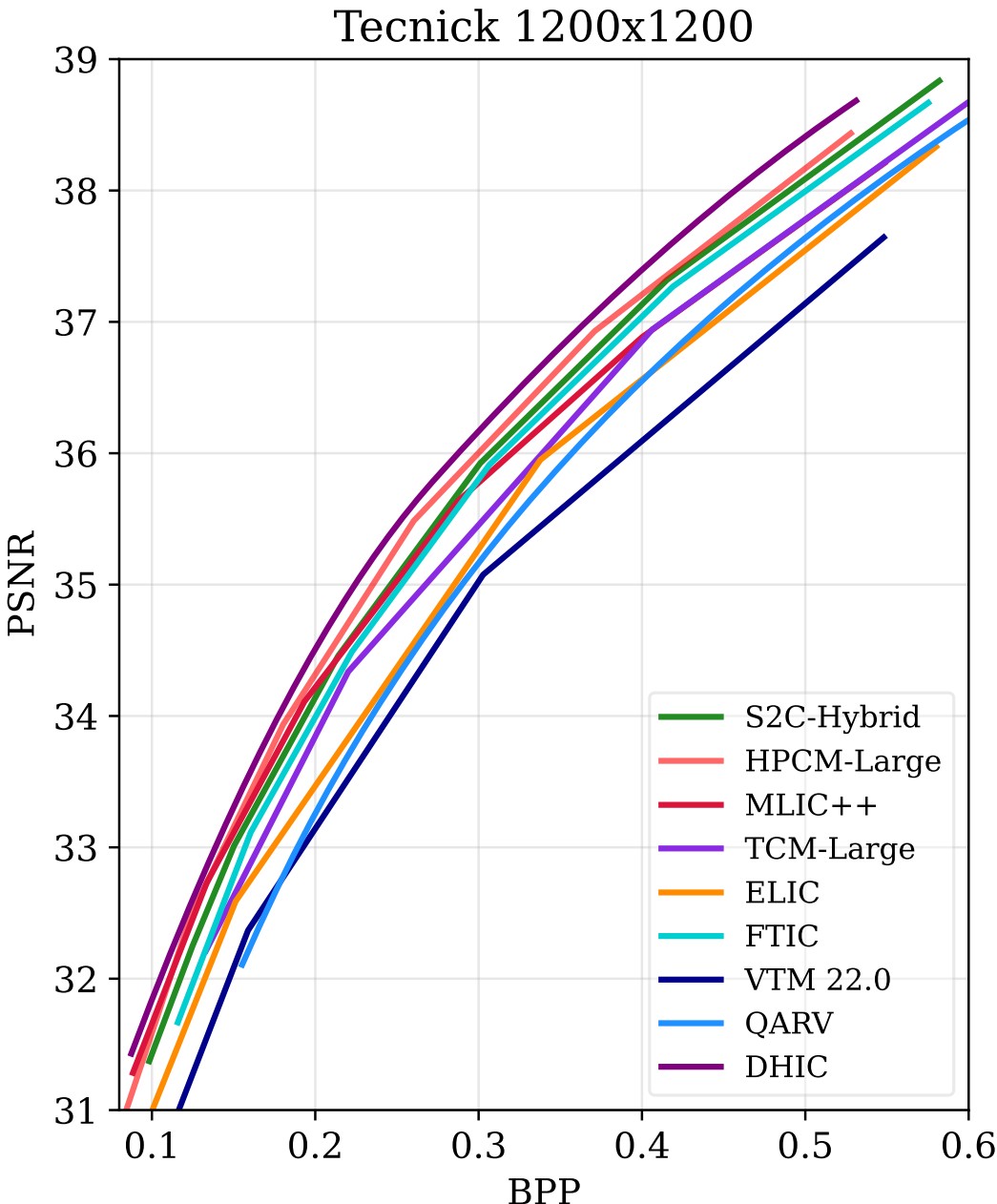

Figure 12: Rate-Distortion curves on Tecnick dataset, all models are optimized with MSE.

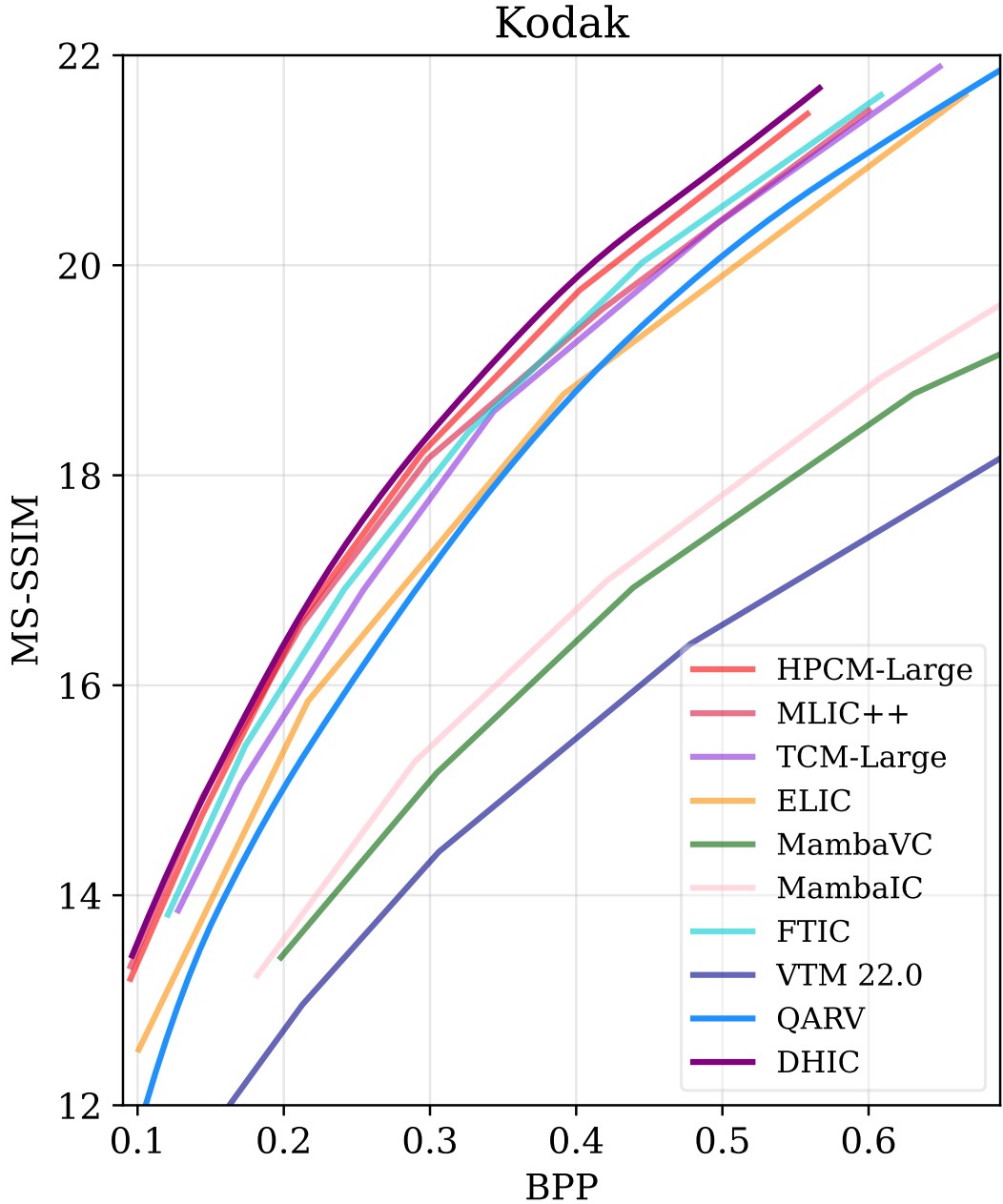

Figure 13: Rate-Distortion curves on Kodak dataset, all models are optimized with MS-SSIM.

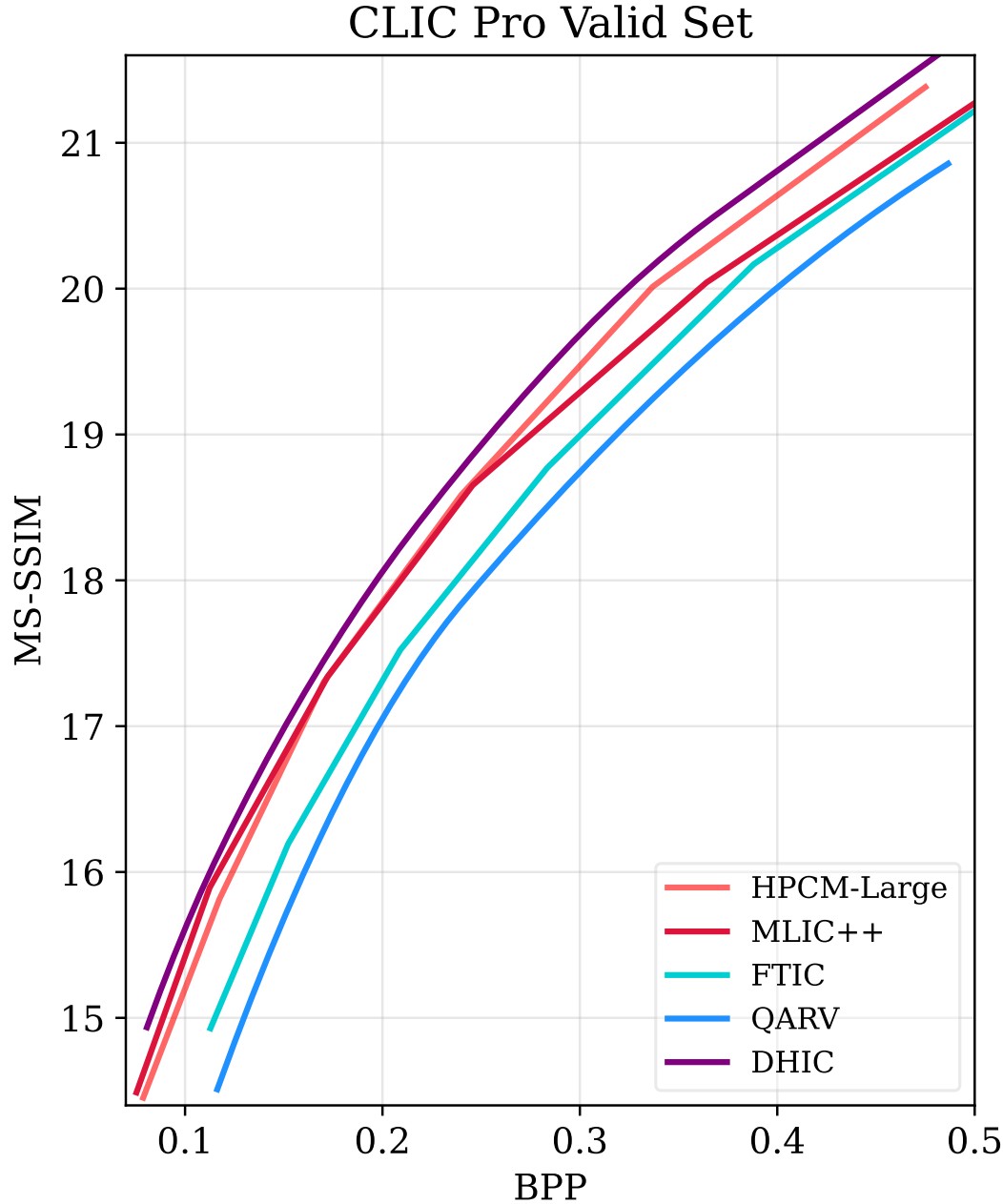

Figure 14: Rate-Distortion curves on CLIC Professional Valid dataset, all models are optimized with MS-SSIM.

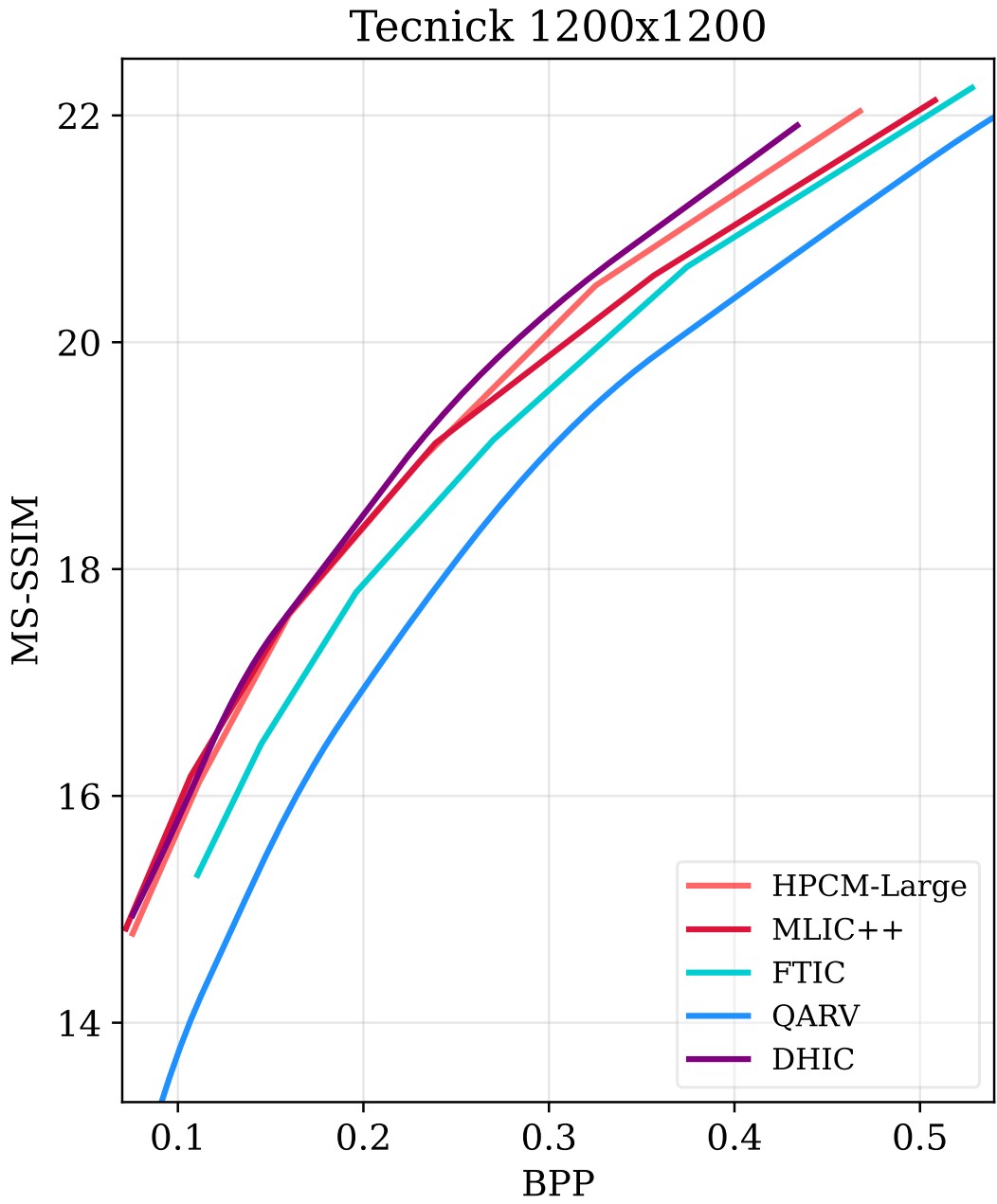

Figure 15: Rate-Distortion curves on Tecnick dataset, all models are optimized with MS-SSIM.

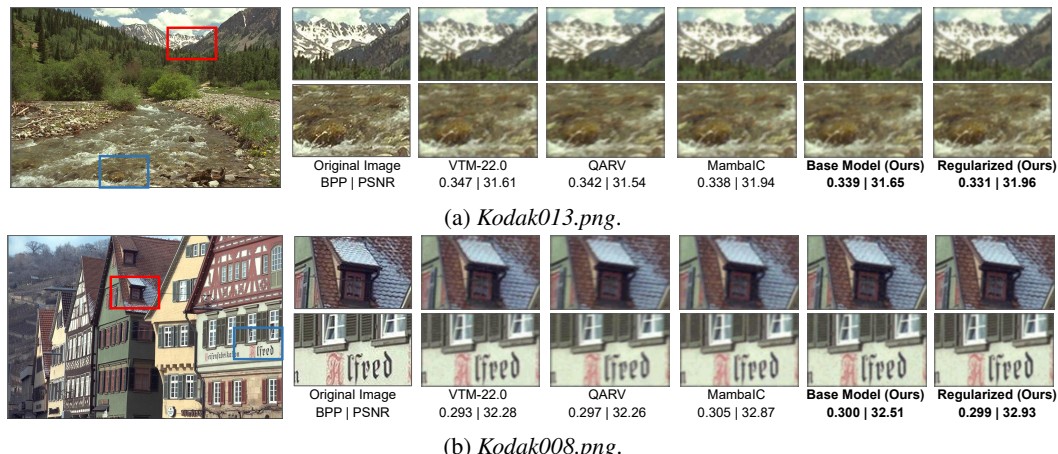

(a) *Kodak013.png*.

(b) *Kodak008.png*.

Figure 16: Subjective quality visualization of decoded reconstruction of various image codecs (Zoom in for more details).

### A.4    MORE ANALYSIS

In this section, we supplement more analysis and experiments of our regularization method to investigate the principles and effectiveness of it. Besides, based on this, we further explore the fundamental advantages and characteristics of hierarchical image codec, especially with regularized optimization, compared to the vanilla hierarchical model with naive training approach and the best single-scale paradigm (e.g., HPCM-Large (Li et al., 2025c)). The details are presented as follows.

**Ablation study on the basic network design:** Compared to the previous hierarchical coding architectures (Duan et al., 2023b), our basic model has integrated some module improvements to further enhance its performance as shown in Fig. 9, including introducing a skip connection like `FSP` module between scales and using reparameterization design in the `Basicblock`. To further quantify their effects, we have supplemented relevant ablation studies here. Moreover, it is worth emphasizing that these designs are only aimed at improving the basic structure, which are not related to the main contribution of this paper—the spectral analysis and regularization training of hierarchical architectures.

The `FSP` module is consist of a DWT module, followed by a `conv 1x1` operation and finally transformed back to feature domain via an IDWT, serving as a skip-like component between scales in our hierarchical coding architecture. In practice, we integrate the `FSP` module only during the finetuning stage after the main regularization training, and get about 0.68% bitrate savings. However, further experiments reveal that this gain primarily resulted from more complete finetuning under an adjusted LR scheduler setup (i.e., `ReduceLROnPlateau` patience increase from 2 to 5), not from the `FSP` itself. Equivalent finetuning schedules without `FSP` can still produce similar rate-distortion gains, and adding `FSP` do not produce noticeable training speedups, too. Therefore, `FSP` is not necessary for our hierarchical architecture. The ablation results are reported in Table 4

Table 4: Ablation study on the `FSP` module. (Baseline anchor for BD-Rate: VTM-22.0, Test dataset: Kodak)

| Metrics | w/o **FSP**+wrong setup | w/ **FSP**+full training | w/o **FSP**+full training |
|---|---|---|---|
| BD-Rate (%) | -19.05 | -19.73 | -19.70 |
| Numbers of finetuning epochs (%) | 46 | 78 | 74 |

Besides, we re-parameterize the original `DWconv 3x3` of `BasicBlock` into a three-branch structure, `DWconv 3×3`, `DWconv 1×1`, `and identity` during training. At test time these branches are fused into a single convolution. This training-time multi-branch design effectively expands model capacity. To quantify its effect, we re-train a network without this re-parameterization and compare R-D performance and training speed; the results are summarized in Table 5.

Table 5: Ablation study on the re-parameterization (abbreviated as Rep.) design. (Baseline anchor for BD-Rate: VTM-22.0, Test dataset: Kodak)

| Metrics | w/o Rep. | w/ Rep. | w/Rep. but keep similar KMACs |
|---|---|---|---|
| BD-Rate (%) | -16.45 | -19.73 | -18.10 |
| Numbers of full training epochs (%) | 350 | 380 | 500 |

**Why L2 distance is preferable in inter-scale regularization:** In general, the gaussian distribution assumption used by the L2 loss aligns more closely with the conditional probability modeling process $p(z_l \mid z_{l-1})$ under the gaussian approximate prior in the hierarchical structure. Using the L2 loss can directly and effectively minimize the negative log-likelihood between $z_{l-1}$ and $z_l$ (i.e., maximize the log-likelihood), forming a more effective mechanism in which the preceding layer predicts the subsequent one.

The detailed derivation is as follows. We already know that in our hierarchical coding architecture, the distribution of the prior $p(z_{1:L})$ can be expressed as

$$p(z_{1:L}) = p(z_0) \prod_{l=1}^{L} p(z_l|z_{l-1}), \tag{7}$$

where $p(z_0)$ is the probability distribution of the initial learnable bias and can be temporarily ignored, while the conditional distribution $p(z_l \mid z_{l-1})$ is typically regarded as a gaussian distribution of the form $\mathcal{N}(f(z_{l-1}), \tau^2 \mathbb{I})$ in hierarchical coding or HVAE.

Its negative log-likelihood can be written as:

$$-\log p(z_l|z_{l-1}) = \frac{1}{2\tau^2}||z_l - f(z_{l-1})||^2 + C, \tag{8}$$

where the first term is exactly an L2 loss form. Therefore, the L2 loss is maximizing the negative log-likelihood between the two latent layers $z_{l-1}$ and $z_l$, meaning that $z_{l-1}$ can better help predict $z_l$. Note that our final objective is to maximize the L2 loss, namely to avoid excessive similarity between the two latent scales, which would lead to spectral aliasing and redundant bitrate. In addition, the ablation study in Table 3 also empirically demonstrates that the alignment effect of the L2 loss is superior to other methods (such as L1 loss or cosine similarity).

**Ablation study on different choices of the weight parameter $\delta$:** We conduct a series of training with different values of $\delta$ to explore the optimal weight parameter. We find that setting $\delta$ too large or too small leads to a decline in rate–distortion performance; therefore, we ultimately set $\delta = 0.1$, as detailed in Table 6.

Table 6: Ablation study on different choices of the weight parameter $\delta$ (Baseline: Our proposed hierarchical architecture without integrating regularization).

| Values of $\delta$ | BD-Rate (%) |
|---|---|
| 0.05 | -7.15 |
| 0.1 | -11.50 |
| 0.2 | -8.68 |

**Compression performance with respect to image resolution:** Test images used in the main text have relatively small spatial resolutions. Yet, high-definition or even ultra-high-definition images are increasingly prevalent in our daily lives, as imaging devices have advanced significantly in recent years. We thus conduct ablation studies to evaluate the compression performance with high-definition images. We compared the performance and complexity of the proposed hierarchical coding model with regularization ("DHIC-Regu"), the base model without regularization ("DHIC-Base"), as well as the latest single-scale model, HPCM-Large (Li et al., 2025c).

In practice, we utilized the LIU4K-v2 valid dataset (Liu et al., 2020), which comprises many high-resolution, complex, and visually high-quality 4K-resolution images. Samples are downsampled

multiple times to a testing dataset with images at different resolutions. We then test the rate-distortion performance, decoding time changes, as shown in Table 7.

As resolution increases, the hierarchical coding structure yields larger performance gains while its decoding time grows more slowly than that of the single-scale baseline. With additional regularization, decoding speed remains effectively constant and performance improves further. These results highlight the hierarchical codec's scaling efficiency, making it well-suited to emerging real-time, high-resolution applications.

Table 7: Compression performance and decoding complexity for images with various resolutions (Baseline anchor for BD-Rate: VTM-22.0). The best method at each resolution is marked in blue.

| Resolution | Method | BD-Rate (%) | Decoding Time (ms) |
|---|---|---|---|
| 480×270 | DHIC-Base | -3.27 | 56.59 |
| | DHIC-Regu | -3.98 | 56.59 |
| | HPCM-Large | -4.12 | 72.80 |
| 960×540 | DHIC-Base | -6.01 | 88.14 |
| | DHIC-Regu | -7.58 | 88.14 |
| | HPCM-Large | -7.50 | 189.92 |
| 1920×1080 | DHIC-Base | -10.65 | 341.18 |
| | DHIC-Regu | -13.98 | 341.18 |
| | HPCM-Large | -10.81 | 648.40 |
| 3840×2160 | DHIC-Base | -13.66 | 1297.44 |
| | DHIC-Regu | -17.19 | 1297.44 |
| | HPCM-Large | -12.98 | 2754.45 |

**Effectiveness of our proposed regularization method at hierarchical codecs with different complexity levels:** Under our current network architecture, by adjusting the network width, depth, and number of cascade modules, we obtain models at different complexity levels; the performance comparison is shown in Table 8. It can be observed that our method consistently yields improvements across models of varying complexity.

Table 8: Effectiveness of our proposed regularization method on hierarchical codecs with different complexity levels (Baseline: VTM-22.0).

| Complexity (KMACs/pix) | BD-Rate (%) | BD-Rate w/regularization (%) | Training Acceleration |
|---|---|---|---|
| 356.41 | 3.38 | -1.74 | 1.8× |
| 683.89 | -1.60 | -11.22 | 2.0× |
| 977.73 | -9.62 | -19.73 | 2.3× |

**Effectiveness of the proposed inter-scale regularization method in other HVAE-based framework:** To validate the generalizability of our proposed regularization method, we implement the regularization on an additional, representative hierarchical coding architecture - QARV (Duan et al., 2023a) and conduct the same experiments. Compared to the hierarchical architecture proposed and used in the main text, QARV shares a similar pipeline but utilizes more latent blocks in the entropy model pathway for each scale. As illustrated in Table 9, the results show that our regularization strategy remains effective on QARV, yielding an approximately 1.65× training speed-up and an 8.20% bitrate savings.

Table 9: Effectiveness of using proposed regularization on QARV (Duan et al., 2023a) (Baseline: vanilla QARV).

| Regularization Setup | BD-Rate (%) | Training Speed |
|---|---|---|
| w/ intra scale | -0.42 | 1.53× |
| w/ inter scale | -6.06 | 1.02× |
| w/ both | -8.20 | 1.65× |

**Applicability to single-scale models with complex context modeling:** Although the proposed regularization method is designed to address the spectral issues observed on multi-scale latents in hierarchical structures. But further, we are also curious *whether this approach is equally effective on different slices in the context modeling process of a single-scale structure*. To this end, we conduct experiments on three representative single-scale codecs with different advanced contexts, ELIC (He et al., 2022), MLIC++ (Jiang et al., 2025), and HPCM (Li et al., 2025c). The results are illustrated in Table 10 below.

Table 10: Further results of integrating the proposed regularization methods (abbreviated as Regu.) on three single-scale codec. (Baseline anchor for BD-Rate: VTM-22.0, Test dataset: Kodak)

| Codecs | Metrics | w/o Regu. | w/ Regu. |
|--------|---------|-----------|----------|
| ELIC | BD-Rate (%) | -3.56 | -3.88 |
|  | Number of epochs | 220 | 205 |
| MLIC++ | BD-Rate (%) | -9.22 | -2.56 |
|  | Number of epochs | 525 | 490 |
| HPCM | BD-Rate (%) | -15.55 | -7.67 |
|  | Number of epochs | 475 | 490 |

It can be seen that integrating the proposed regularization on ELIC (He et al., 2022) can bring slight performance and training speed gains, but it is not as obvious as our hierarchical structure. In the context modeling processes of MLIC++ (Jiang et al., 2025) and HPCM (Li et al., 2025c), which are more complex, performance degradation even occurs. This suggests that additional customized designs may be needed to adapt to single-scale VAEs, which is also a promising direction for our future work.

In depth, We analyze the essential differences between hierarchical coding structures and single-scale ones with various context modeling process.

Hierarchical models perform explicit scale-by-scale transformations that naturally produce multi-scale latents, where each scale roughly corresponds to a specific frequency band. Finally, it can conduct scale-wise coding process by using upper-scale latent as a condition to assist in modeling lower-scale one. Based on this, integrating the proposed intra-scale and inter-scale explicit regularization into this process, essentially corresponds scale-by-scale to the entire pipeline of model feature transformation, latent modeling, and decoding reconstruction, making it easier to address issues such as spectral energy dissipation within scales and spectral aliasing between scales.

By contrast, for single-scale models with various context modeling, they rely on powerful conditional modeling capabilities to directly fit the single-scale latent conditional distribution. Essentially, it is still a conditional probability modeling of the single-scale latent obtained from a single-scale transformation (the hierarchical design in HPCM Li et al. (2025c) is also a hierarchical conditional probability modeling of the single-scale latent). There is no explicit multi-scale transformation of the input signal corresponding to the multi-scale latent design, so it cannot guarantee the natural frequency decomposition process. In other words, in the context modeling process, it is difficult to ensure that different slice components of a single-scale latent can be effectively decomposed on the spectrum and modeled scale by scale. Blindly using such regularization may even disrupt the original context modeling design.

## B    THE USE OF LARGE LANGUAGE MODELS

We would like to acknowledge the assistance of AI tools in improving the expression of partial writing in this paper. All ideas, methods, code implementations, experiments, and the overall conceptualization of this work were independently developed by the authors, with no involvement of AI.

