# OpenReview forum: "Taming Hierarchical Image Coding Optimization: A Spectral Regularization Perspective"
_ICLR.cc/2026/Conference — ICLR 2026 Poster_

### Official Review · Reviewer_RfuU · 2025-10-24

**Soundness:** 3
**Presentation:** 2
**Contribution:** 3
**Rating:** 4
**Confidence:** 4

**Summary:**

They propose explicit spectral regularization (intra-scale frequency regularization for smooth frequency buildup across scales, inter-scale similarity regularization to suppress cross-scale aliasing) — applied only in training with no inference overhead. Experiments show the method accelerates vanilla model training by 2.3x, achieves 20.65% average rate–distortion gain over VTM-22.0 on public datasets, outperforms single-scale approaches, and sets a new SOTA in learned image compression.

**Strengths:**

1. This manuscript tries to extract disentangled features from images using spectral regularization.

2. The motivation is clear.

3. The results about the performance of the proposed method are convincing in Kodak, CLIC datasets.

**Weaknesses:**

1. The related details of the proposed method are not described clearly. For example, how to obtain the y-axis of Figure 1 ? What does it mean about the lines with different color of Figure 8?

2. The training of inter-scale regularization as shown in Figure 5 may not be stable since $z_1, z_2$ maybe the noise at the early stage and the corresponding regularization makes no sense.

3. There exists no analysis or discussion about "Different scales converge to their respective frequency bands at different rates" at line 183. It could be better if this manuscript provides some clues.

4. For visualization in Figure 6, it is hard to totally distinguish the frequencies of the details without any aliasing. Besides the demonstration of Figure 6(b) has little color shift compared with 6(c), can the regularization be more effective for color representation?

**Questions:**

See Weaknesses.

---

> ### Author Response · Authors · 2025-11-20
> **Response to Reviewer RfuU Part 1**
>
> Thank you for the thorough review and for acknowledging the clear motivation and strong experimental performance. We address your concerns below.
>
> **W1: Details for proposed method (e.g., Figure 1 and Figure 8)**
>
> We supplement more implementation details about our method in the revisited manuscript Sec.3 and Appendix, as well as the detailed introduction of Figure 1 and Figure 8, including:
>
> - **Further explanation of the inter-scale regularization:** We believe that intra-scale regularization is relatively simple and easy to understand, with only high-frequency truncation operations gradually released with the number of epochs, to form a learning pattern from low frequency to high frequency. We focus on further supplementing the principles and implementation details of inter-scale regularization.
>
>   we clarify that the inter-scale regularization is designed to *minimize* latent similarity across scales. There is a small typo in Eq. (6): the regularization term should be preceded by a “−” (minus) sign rather than a “+”. We apologize for the ambiguity. We have corrected it in the revised manuscript and supplemented relative explanation in Sec. 3.3 and Appendix A.2.
>
>   Next, we elaborate on the motivation and mechanism:
>
>   - **Problem setting.** hierarchical coding  is designed to progressively refine representations across scales[1] [2]: the latent from a coarser scale is used as a condition to predict the next finer scale. Often the architecture even explicitly uses residual addition between scales (see Fig. 9 latent block). In image signals, such residuals commonly correspond to higher-frequency details, while shallower latents capture lower-frequency structure (consistent with classical predictive / wavelet coding phenomena[3]). However, simple one-shot synchronous optimization across all scales lacks an explicit mechanism that ensures this intended sequential residual learning, producing redundancy across scales and manifesting as the spectral aliasing observed in Fig. 1.
>   - **Principle.** The inter-scale regularization aims to align the coarser-scale latent to the subspace of the finer-scale latent (via a frequency-aware transform) and then penalize predictable (i.e., similar) components. Intuitively, if a component of a finer-scale latent can be well predicted from the previous scale, it is likely a low-frequency component that can be better represented at the coarser scale. Penalizing this predictable similarity removes those low-frequency components from the finer latent and leaves it to capture higher-frequency, less predictable residuals—thereby mitigating cross-scale spectral aliasing.
>   - **Implementation.** We perform an explicit frequency-aware alignment by applying a discrete wavelet (DWT)-based transform to the coarser latent, which decomposes it into frequency sub-bands. We then apply a 1×1 convolution across channels to linearly map and recompose frequency sub-bands so they align with the finer-scale latent’s frequency channels. The inter-scale penalty then discourages the aligned coarser latent from predicting the same low-frequency content in the finer latent. Thus, the finer-scale latent retains the difficult-to-predict high-frequency components while redundant low-frequency parts are suppressed, alleviating spectral aliasing.
> *W1's response is to be continued*

---

> ### Author Response · Authors · 2025-11-20
> **Response to Reviewer RfuU Part 2**
>
> *Starting with the remaining part of the reply to W1*
>
> - **The acquisition method of y-axis and more Implement Details of Figure 1:** Figure 1 is a heatmap that measures the degree of spectral overlap between the scale-wise latent and the original image during training (in practice, this is reflected through the progressive decoded reconstruction of each scale). The horizontal axis represents the number of training epochs, and the vertical axis represents the normalized frequency range. A point at (epoch, frequency position) indicates the spectral-overlap intensity between the latent of a certain scale and the original image at that frequency when training reaches the current epoch. The specific implementation steps are as follows:
>   - Using progressive decoding (the implementation is detailed in Appendix A.1), obtain $\hat{x}_l$, which is normally decoded up to layer l, while the remaining $L-l$ layers directly use the mean value of prior.
>   - Define $I_l = \hat{x}_l - \hat{x}_{l-1}, (I_0 = \hat{x}_0)$ as the mutual information between the reconstruction using only the latent of scale l and the original image.
>   - Apply a 2D-DCT transform to the original image $x$ and each layer’s mutual information $I_l$ to obtain their spectra $F_x$ and $F_l$.
>   - Perform radial binning on the 2D spectrum and convert it into a 1D form.
>   - Normalize the frequency range and compute the spectral-overlap degree between the mutual information $I_l$ and the original image $x$ in each bin: $E = \sum_{b_i}^{b_{i+1}} \frac{F_l^{b_i}}{F_x^{b_i}}$.
>   - Finally, use the matplotlib plotting script to generate the heatmap for each scale.
> - We have added the implement details and matplotlib plotting script for Figure 1 in Appendix A.1. To ensure full reproducibility, we will open-source the code and additional related plotting scripts after acceptance.
> - **The meaning of different color in Figure 8:** In Fig.8, different sub-graphs have different colors representing different scales. In the same sub-graph, light colors represent training without any regularization, while dark colors represent training with the proposed regularization.
>
> **W2: Stability of inter-scale regularization at the early stage of training**
>
> This is a valid concern, but in fact we have already taken this issue into account (perhaps our description in the paper was not prominent enough, causing reviewers to overlook this point; in the revised version we have further clarified when we practically apply the inter-scale regularization).
>
> At the early stage of training, we first integrate only the intra-scale regularization (for about 100 epochs) to ensure that the latent of each scale can quickly and stably converge to its appropriate frequency band. After this, we then switch to the inter-scale regularization to eliminate spectral aliasing among different scales. We also attempted applying both intra-scale regularization and inter-scale regularization from the beginning, and indeed found that it leads to obvious fluctuations in convergence and bottlenecks in rate–distortion performance.

---

> ### Author Response · Authors · 2025-11-20
> **Response to Reviewer RfuU Part 3**
>
> **W3: “Different scales converge at different rates” — evidence and discussion**
>
> Thank you for your constructive comment. This conclusion is primarily derived from studies on spectral bias/spectral principles; we have discussed this phenomenon in Sec. 2.2 of the paper in the context of hierarchical structures, and our existing experiments can also support the conclusion. For example, Figure 1 shows that scale1 has essentially converged to its corresponding spectral position (and no longer increases) at around epoch 30; subsequently, scale2 converges at around epoch 50, scale3 at around epoch 80, and scale4 at around epoch 100. The original training loss curves in Figure 8 also roughly follow this trend, which validates our claim that “different scales converge to their respective frequency bands at different rates.”
>
> In addition, we have supplemented more prior-work discussion related to this point in the revisited version Appendix A.1, specifically including:
>
> 1. Studies on neural network spectral bias / spectral principles [4] [5] [6] have shown that networks exhibit a preference for low-frequency functions during training; low frequencies are learned earlier and faster than high frequencies in deeper module of the network. Mapping this finding to our network structure can explain why different scales exhibit different convergence speeds for different frequency bands.
> 2. Existing HVAE and hierarchical coding works [1] [2] [3] [7] indicate that different hierarchical scales tend to capture information at different frequencies: low-resolution scales capture more abstract global information (typically low-frequency structures), whereas high-resolution scales handle local details and residual information (typically high-frequency).
>
> Taken together, these theoretical analysis and visualizations demonstrate, in our hierarchical structure, different scales will converge to different frequency bands at different rates: deeper, lower-resolution scales converge faster to low-frequency regions, and shallower, higher-resolution scales then gradually converge to their corresponding high-frequency positions on that basis.

---

> ### Author Response · Authors · 2025-11-20
> **Response to Reviewer RfuU Part 4**
>
> **W4: Further explanation of the Figure 6 visualizations**
>
> Figure 6 compares latent reconstructions across training, demonstrating that with our regularization scales disentangle earlier and more clearly. The observed color differences in the visualization arise from the colormap scaling used for display (i.e., differences in latent magnitude ranges), not from inherent color-representation artifacts of the regularizer. We clarified this in the revised caption and text and improved the visualization for clarity.
>
> Thanks again for the detailed review. We hope these clarifications are helpful and would appreciate your reconsideration of the score.
>
> Figure 6 shows a fairly clear contrast between the latents of each scale obtained with and without regularized training. We expand the explanation below and have revised the statement in the latest manuscript to improve readability and clarity.
>
> Sub-figure 6(a) presents the test example *kodak002.png*: an image of a door that contains both low-frequency content (large uniform color regions and smooth horizontal/vertical structures) and high-frequency details (the latch, fine wood grain, etc.). Sub-figures (b) and (c) correspond to the original (unregularized) training and the regularized training, respectively. In both sub-figures, the horizontal axis runs to the right with increasing training epochs, and the vertical axis runs downward from low-resolution, deeper scales to high-resolution, shallower scales. Comparing the two trainings, we observe:
>
> 1. During the original training, the latents across different scales remain entangled and hard to separate throughout training. Deeper scales (e.g., S1) fail to capture global semantics, while shallower scales (e.g., S4) cannot effectively represent high-frequency structures and instead exhibit disorderly noisy patterns. The model also produces grid-like artifacts and central blurring; only after around epoch 120 do some structured features start to appear, and even then clear scale decoupling is not achieved.
> 2. By contrast, in regularized training, well-defined latents emerge as early as epoch 40, and by roughly epoch 120 to 160 the characteristics of each scale are already quite distinct. Subsequently, features at each scale continue to be refined and naturally decouple, ultimately forming a clear coarse-to-fine hierarchical information structure.
>
> This comparison naturally and intuitively demonstrates that our intra-scale regularization ensures each scale rapidly and accurately converges to its own frequency band in the early training stage—thereby preventing energy dissipation—while the inter-scale regularization applied later helps mitigate spectral aliasing between scales.
>
> Regarding the reviewer’s comment about color differences: this is likely caused by the colormap used when plotting, which is unrelated to the inherent color representation of the latents themselves.
>
> We appreciate your time and constructive feedback. Hope our clarifications can resolve your issues. If you find the response satisfactory, we would be grateful for a higher score.
>
> **Refs:**
>
> [1] Sønderby C K, Raiko T, Maaløe L, et al. Ladder variational autoencoders[J]. Advances in neural information processing systems, 2016, 29.
>
> [2] Vahdat A, Kautz J. NVAE: A deep hierarchical variational autoencoder[J]. Advances in neural information processing systems, 2020, 33: 19667-19679.
>
> [3] Duan Z, Lu M, Ma J, et al. Qarv: Quantization-aware resnet vae for lossy image compression[J]. IEEE Transactions on Pattern Analysis and Machine Intelligence, 2023, 46(1): 436-450.
>
> [4] Rahaman N, Baratin A, Arpit D, et al. On the spectral bias of neural networks[C]//International conference on machine learning. PMLR, 2019: 5301-5310.
>
> [5] Xu Z Q J, Zhang Y, Luo T, et al. Frequency principle: Fourier analysis sheds light on deep neural networks[J]. arXiv preprint arXiv:1901.06523, 2019.
>
> [6] Xu Z J, Zhou H. Deep frequency principle towards understanding why deeper learning is faster[C]//Proceedings of the AAAI conference on artificial intelligence. 2021, 35(12): 10541-10550.
>
> [7] Duan Z, Lu M, Ma Z, et al. Lossy image compression with quantized hierarchical vaes[C]//Proceedings of the IEEE/CVF winter conference on applications of computer vision. 2023: 198-207.

---

> ### Comment · Reviewer_RfuU · 2025-11-23
> **Response of authors**
>
> Thanks of the detailed explanation and the response of stability of inter-scale regularization can convince me, thus I will improve my rating. Though I still doubt about the discussion of spectral bias / spectral principles for this work, and maybe a more effective theory need deeper research.

---

> > ### Author Response · Authors · 2025-11-25
> >
> > Thank you very much for your careful reading and for being willing to raise your rating — we greatly appreciate it. We also appreciate your concern about the spectral-bias / frequency-principle discussion. We agree that a deeper theoretical treatment would strengthen our work. To address this, we further improve our manuscript, in the latest revised version Appendix A.1 we have:
> > 1. Further reviewed the research on Prior Spectral Bias/Frequency Principle.
> > 2. Expanded our discussion section to explicitly connect our empirical observations to established work on the Frequency Principle / spectral bias.

---

### Official Review · Reviewer_HdPe · 2025-10-29

**Soundness:** 4
**Presentation:** 3
**Contribution:** 3
**Rating:** 6
**Confidence:** 3

**Summary:**

The authors proposed two techniques to address the inefficiency in training in hierarchical coding. The first is for intra-scale frequency regularization, which is to guide the training frequencies by progressive truncation. The second for inter-scale similarity regularization, which is to suppress the similarity between neighboring scales (?). Experimental results demonstrate the successfulness of the proposed method.

For the weakness, 1. The inter-scale similarity regularization part needs more explanation. It looks like that Eq.(6) encourages similarity instead of suppressing it. 2. Some figures are not helpful for explaining the method, e.g. Figs 4 and 5.

**Strengths:**

The authors proposed two techniques to address the inefficiency in training in hierarchical coding. The first is for intra-scale frequency regularization, which is to guide the training frequencies by progressive truncation. The second for inter-scale similarity regularization, which is to suppress the similarity between neighboring scales (?). Experimental results demonstrate the successfulness of the proposed method.

**Weaknesses:**

1. The inter-scale similarity regularization part needs more explanation. It looks like that Eq.(6) encourages similarity instead of suppressing it.
2. Some figures are not helpful for explaining the method, e.g. Figs 4 and 5.

**Questions:**

1.	The inter-scale similarity regularization part needs more explanation. It looks like that Eq.(6) encourages similarity instead of suppressing it.
2.	Some figures are not helpful for explaining the method, e.g. Figs 4 and 5.
3.	What is the meaning of “time t”?
4.	The inner minimum in Eq.(4) seems not necessary.

---

> ### Author Response · Authors · 2025-11-20
> **Response to Reviewer HdPe**
>
> Thank you for the careful review and for **confirming the effectiveness of our two proposed techniques**. Below we respond to your points.
>
> **W1 & Q1: Inter-scale regularization and Eq. (6) typo**
>
> We clarify that the inter-scale regularization is designed to *minimize* latent similarity across scales. There is a small typo in Eq. (6): the regularization term should be preceded by a “−” (minus) sign rather than a “+”. We apologize for the ambiguity. We have corrected it in the revised manuscript and supplemented relative explanation in Sec. 3.3 and Appendix A.1.
>
> Next, we elaborate on the motivation and mechanism:
>
> - **Problem setting.** hierarchical coding  is designed to progressively refine representations across scales[1] [2]: the latent from a coarser scale is used as a condition to predict the next finer scale. Often the architecture even explicitly uses residual addition between scales (see Fig. 9 latent block). In image signals, such residuals commonly correspond to higher-frequency details, while shallower latents capture lower-frequency structure (consistent with classical predictive / wavelet coding phenomena[3]). However, simple one-shot synchronous optimization across all scales lacks an explicit mechanism that ensures this intended sequential residual learning, producing redundancy across scales and manifesting as the spectral aliasing observed in Fig. 1.
> - **Principle.** The inter-scale regularization aims to align the coarser-scale latent to the subspace of the finer-scale latent (via a frequency-aware transform) and then penalize predictable (i.e., similar) components. Intuitively, if a component of a finer-scale latent can be well predicted from the previous scale, it is likely a low-frequency component that can be better represented at the coarser scale. Penalizing this predictable similarity removes those low-frequency components from the finer latent and leaves it to capture higher-frequency, less predictable residuals—thereby mitigating cross-scale spectral aliasing.
> - **Implementation.** We perform an explicit frequency-aware alignment by applying a discrete wavelet (DWT)-based transform to the coarser latent, which decomposes it into frequency sub-bands. We then apply a 1×1 convolution across channels to linearly map and recompose frequency sub-bands so they align with the finer-scale latent’s frequency channels. The inter-scale penalty then discourages the aligned coarser latent from predicting the same low-frequency content in the finer latent. Thus, the finer-scale latent retains the difficult-to-predict high-frequency components while redundant low-frequency parts are suppressed, alleviating spectral aliasing.
>
> **W2 & Q2: Figures 4 and 5 readability**
>
> We revised Figures 4 and 5 in the updated manuscript to improve readability and help convey the method more clearly.
>
> **Q3: “time t” in Eq. (4)**
>
> `t` denotes training epoch index. We added this clarification in the revised manuscript.
>
> **Q4: Inner minimum in Eq. (4)**
>
> Thank you for your suggestion. The inner `min` in Eq. (4) is unnecessary (the latter term is always less than 1). We removed it in the revisited revision.
>
> We appreciate your constructive suggestions and hope our clarifications resolve your concerns. If you find the updates satisfactory, we would be grateful for a higher score.
>
> **Refs:**
>
> [1] Sønderby C K, Raiko T, Maaløe L, et al. Ladder variational autoencoders[J]. Advances in neural information processing systems, 2016, 29.
>
> [2] Vahdat A, Kautz J. NVAE: A deep hierarchical variational autoencoder[J]. Advances in neural information processing systems, 2020, 33: 19667-19679.
>
> [3] Duan Z, Lu M, Ma J, et al. Qarv: Quantization-aware resnet vae for lossy image compression[J]. IEEE Transactions on Pattern Analysis and Machine Intelligence, 2023, 46(1): 436-450.

---

### Official Review · Reviewer_QW9U · 2025-10-30

**Soundness:** 2
**Presentation:** 4
**Contribution:** 3
**Rating:** 6
**Confidence:** 5

**Summary:**

This paper tackles optimization bottlenecks in hierarchical image compression by identifying that standard training suffers from "spectral aliasing" and "energy dispersion" across scales. To resolve this, the authors introduce two training-only spectral regularizers: an intra-scale DCT-based curriculum for progressive low-to-high frequency learning, and an inter-scale similarity penalty to reduce redundancy. This approach accelerates training by 2.3x and sets a new state-of-the-art, achieving a 20.65% average BD-Rate saving over the VTM-22.0 video codec. I like the paper's insightful analysis and strong empirical results. However, its methodological soundness is weakened by a lack of theoretical justification connecting the proposed regularization schemes to their stated goal of spectral separation.

**Strengths:**

1.The paper provides a novel and insightful spectral analysis, identifying "energy dispersion" and "spectral aliasing" as the root causes of training difficulties in hierarchical compression models. The compelling visualizations strongly support this diagnosis.
2.The proposed regularization techniques are well-designed and directly target the diagnosed spectral issues. The approach is rigorous, logically sound, and adds no inference overhead, making it practical.
3.The method achieves state-of-the-art performance with substantial gains, including a 20.65% average BD-Rate saving over VTM-22.0. The 2.3x training acceleration is a significant practical advantage.
4.The paper is supported by thorough experiments, including extensive SOTA comparisons, detailed ablation studies, and proven generalizability to other architectures, which robustly validate the authors' claims.

**Weaknesses:**

1.The paper's design, guided by the "frequency principle," assigns low-frequency content to semantically deeper scales and high-frequency content to shallower scales. This creates an apparent tension with the common understanding that more challenging, high-frequency details often benefit from the greater expressive power of deeper network layers. While the empirical results are strong, the paper's methodological contribution would be further strengthened by a clearer theoretical justification for this seemingly counter-traditional design, explaining why assigning the most complex information to relatively shallower structures is advantageous in this context.

2.The paper lacks a theoretical basis for its claim that an L2 latent penalty (Eq. 6) enforces spectral separation between scales. The connection between latent distance and spectral orthogonality is not established. Furthermore, the implementation uses a 1x1 convolution, a feature-mixing operation that could potentially contradict the goal of suppressing spectral aliasing. The authors should clarify the true source of this regularizer's effectiveness.

**Questions:**

1.Regarding Figure 1:
 Could you provide a more detailed technical explanation of how these heatmaps were generated? Specifically, how is the "spectral overlap" between a scale's contribution and the input image quantitatively defined and computed? The current description in the Appendix is a bit brief for full reproducibility.

2.Regarding Figure 9:
Could you elaborate on the specific design purpose and function of the FSP?
What is the relationship between the FSP and the proposed Inter-Scale Latent Regularization? Do they work synergistically, or are they independent components?
I noticed a similar shortcut structure in a recent related work, AuxT [R1]. Could you clarify the key differences and relationships between your FSP and the auxiliary structure in AuxT?

[R1] Li, et. al, On Disentangled Training for Nonlinear Transform in Learned Image Compression. ICLR 2025.

3.Can the proposed spectral regularization methods be adapted for non-hierarchical (single-scale) compression frameworks?

---

> ### Author Response · Authors · 2025-11-20
> **Response to Reviewer QW9U Part 1**
>
> Thank you for your detailed review and for recognizing our “**novel and insightful spectral analysis**” and “**state-of-the-art performance**.” We appreciate your constructive suggestions. Below we respond to your points.
>
> **W1: On assigning low/high frequency to different scales and network depth**
>
> We believe this concern arises from a conceptual conflation. It is important to distinguish between (i) the representational power of deeper network layers (deeper layers can learn more complex abstractions) and (ii) the mapping rules between hierarchical *scales* and *frequency bands* in hierarchical coding architecture. These two statements do not contradict each other. Specifically:
>
> - The former aims to express that neural networks often require deeper network structures for stronger feature expression capabilities. In this case, deep networks often extract more abstract semantic information from input signals (note that this semantic information does not directly correspond to high-frequency details, but rather a global high-dimensional expression)
> - While the latter statement illustrates that in the hierarchical coding structure, signals are decomposed into multi-scale latent expressions, where deeper, lower resolution scales capture some global information (usually low-frequency structures), while shallower, high-resolution scales capture some residual information (usually high-frequency details). This is in line with the characteristics of hierarchical structure, where deeper, low resolution scale features will serve as conditions to help predict shallower, high-resolution scales, namely low-frequency information helps predict high-frequency information.
> - If there is a conflict between the two statements, that is, the distribution of features at different scales in the hierarchical coding structure is reversed, due to the high information entropy of high-frequency information, it is difficult to fully express it at low resolution scales, and on the other hand, it is unhelpful for predicting low-frequency information, which violates the design of the hierarchical structure. For more detailed theory and introduction from prior works, please refer to the study of HVAE [1] [2] [3] or spectral analysis theory[4] [5].

---

> ### Author Response · Authors · 2025-11-20
> **Response to Reviewer QW9U Part 2**
>
> **W2: Further explanation of inter-scale regularization and Eq. (6)**
>
> Firstly, we clarify that the inter-scale regularization is designed to *minimize* latent similarity across scales. There is a small typo in Eq. (6): the regularization term should be preceded by a “−” sign rather than a “+”. We apologize for the ambiguity. We have corrected it in the revised manuscript and added relative explanation in Sec. 3.3 and Appendix A.1.
>
> Next, we elaborate on the motivation and mechanism:
>
> - **Problem setting.** hierarchical coding  is designed to progressively refine representations across scales[1] [2]: the latent from a coarser scale is used as a condition to predict the next finer scale. Often the architecture even explicitly uses residual addition between scales (see Fig. 9 latent block). In image signals, such residuals commonly correspond to higher-frequency details, while shallower latents capture lower-frequency structure (consistent with classical predictive / wavelet coding phenomena[3]). However, simple one-shot optimization across all scales lacks an explicit mechanism that ensures this intended sequential residual learning, producing redundancy across scales and manifesting as the spectral aliasing observed in Fig. 1.
>
> - **Principle.** The inter-scale regularization aims to align the coarser-scale latent to the subspace of the finer-scale latent (via a frequency-aware transform) and then penalize predictable (i.e., similar) components. Intuitively, if a component of a finer-scale latent can be well predicted from the previous scale, it is likely a low-frequency component that can be better represented at the coarser scale. Penalizing this predictable similarity removes those low-frequency components from the finer latent and leaves it to capture higher-frequency, less predictable residuals—thereby mitigating cross-scale spectral aliasing.
>
> - **Implementation.** We perform an explicit frequency-aware alignment by applying a DWT-based transform to the coarser latent, which decomposes it into frequency sub-bands. We then apply a 1×1 convolution across channels to linearly map and recompose frequency sub-bands so they align with the finer-scale latent’s frequency channels. The inter-scale penalty then discourages the aligned coarser latent from predicting the same low-frequency content in the finer latent. Thus, the finer-scale latent retains the difficult-to-predict high-frequency components while redundant low-frequency parts are suppressed, alleviating spectral aliasing.
>
> - **Why L2 distance is preferable**: In general, the Gaussian distribution assumption used by the L2 loss aligns more closely with the conditional probability modeling process $p(z_l \mid z_{l-1})$ under the Gaussian approximate prior in the hierarchical structure. Using the L2 loss can directly and effectively minimize the negative log-likelihood between $z_{l-1}$ and $z_l$ (i.e., maximize the log-likelihood), forming a more effective mechanism in which the preceding layer predicts the subsequent one.
>
>   The detailed derivation is as follows: We already know that in our hierarchical coding architecture, the distribution of the prior $p(z_{1:L})$ can be expressed as $p(z_{1:L})=p(z_0)\prod_{l=1}^{L} p(z_l|z_{l-1})$.
>
>   where $p(z_0)$ is the probability distribution of the initial learnable bias and can be temporarily ignored, while the conditional distribution $p(z_l \mid z_{l-1})$ is typically regarded as a Gaussian distribution of the form $\mathcal{N}(f(z_{l-1}), \tau^2 \mathbb{I})$ in hierarchical coding or HVAE. Its negative log-likelihood can be written as $-\log p(z_l|z_{l-1})=\frac{1}{2\tau^2}||z_l-f(z_{l-1})||^2+C$.
>
>   where the first term is exactly an L2 loss form. Therefore, the L2 loss is maximizing the negative log-likelihood between the two latent layers $z_{l-1}$ and $z_l$, meaning that $z_{l-1}$ can better help predict $z_l$. Note that our final objective is to maximize the L2 loss, namely to avoid excessive similarity between the two latent layers, which would lead to spectral aliasing and redundant bitrate. In addition, the ablation study in Table 3(b) of this paper also empirically demonstrates that the alignment effect of the L2 loss is superior to other methods (such as L1 loss or cosine similarity).
>
> - **About the 1×1 convolution:** we emphasize that before conv 1x1, we have already decomposed the latent of previous scale by frequency bands through DWT transformation, where the pixel positions represent the intensity of frequency components, and the channel dimension represents different frequency sub-bands and directions. And conv 1x1 only recombines between channels in different frequency bands at the same position, without generating additional frequency aliasing problems through cross position fusion. This operation is only for spectral alignment with the later scale, so it does not conflict with the original intention of suppressing spectral aliasing.

---

> ### Author Response · Authors · 2025-11-20
> **Response to Reviewer QW9U Part 3**
>
> **Q1: Details on Figure 1**
>
> Thank you for your constructive comments. Figure 1 is a heatmap that measures the degree of spectral overlap between the scale-wise latent and the original image during training (in practice, this is reflected through the progressive decoded reconstruction of each scale). The horizontal axis represents the number of training epochs, and the vertical axis represents the normalized frequency range. A point at (epoch, frequency position) indicates the spectral-overlap intensity between the latent of a certain scale and the original image at that frequency when training reaches the current epoch. The specific implementation steps are as follows:
>
> 1. Using progressive decoding (the implementation is detailed in Appendix A.1), obtain $\hat{x}_l$, which is normally decoded up to layer l, while the remaining $L-l$ layers directly use the mean value of prior.
> 2. Define $I_l = \hat{x}_l - \hat{x}_{l-1}, (I_0 = \hat{x}_0)$ as the mutual information between the reconstruction using only the latent of scale l and the original image.
> 3. Apply a 2D-DCT transform to the original image $x$ and each layer’s mutual information $I_l$ to obtain their spectra $F_x$ and $F_l$.
> 4. Perform radial binning on the 2D spectrum and convert it into a 1D form.
> 5. Normalize the frequency range and compute the spectral-overlap degree between the mutual information $I_l$ and the original image $x$ in each bin: $E = \sum_{b_i}^{b_{i+1}} \frac{F_l^{b_i}}{F_x^{b_i}}$.
> 6. Finally, use the matplotlib plotting script to generate the heatmap for each scale.
>
> We have added the implement details and matplotlib plotting script for Figure 1 in Appendix A.1. To ensure full reproducibility, we will open-source the code and additional related plotting scripts after acceptance.
>
> **Q2: FSP module’s role and necessity**
>
> Thanks for this insightful question. Our main contribution focus on the intra-scale and inter-scale regularization, while the FSP module was only used as a skip-like component between scales in our hierarchical coding  architecture. In practice, we added FSP only during a finetuning stage after the main regularization training, and observed ≈0.68% bitrate savings. However, further experiments revealed that this gain primarily resulted from more complete finetuning under an adjusted LR scheduler setup (ReduceLROnPlateau patience increased from 2 to 5), not from FSP itself. Equivalent finetuning schedules without FSP produced similar rate-distortion gains, and adding FSP did not produce noticeable training speedups. Therefore, FSP is not necessary for our hierarchical architecture. The ablation results are reported in Table 1 and supplemented in the revisited version Appendix.A.1 and A.3.
>
> |                                        | w/o FSP + unfull training (wrong setup) | w/ FSP + full training | w/o FSP + full training |
> | -------------------------------------- | --------------------------------------- | ---------------------- | ----------------------- |
> | BD-Rate over VTM-22.0 on Kodak dataset | -19.05                                  | -19.73                 | -19.7                   |
> | Numbers of Epoches in finetune stage   | 46                                      | 78                     | 74                      |

---

> ### Author Response · Authors · 2025-11-20
> **Response to Reviewer QW9U Part 4**
>
> **Q5: Applicability to single-scale models**
>
> This is a valuable suggestion. We had similar thoughts before and performed preliminary experiments integrating our regularization into ELIC[4], MLIC++[5], and HPCM[6]. For ELIC with spatial & channel-wise context modeling, it performs slight training acceleration and RD gain. But for larger models with more complex context (e.g., MLIC++ with multi-reference design, HPCM with hierarchical context design), integrating our regularization instead decreased final performance. We provide both qualitative analysis and quantitative results below.
>
> - **Qualitative analysis:** We first analyzed the essential differences between hierarchical coding  structures and single-scale ones with various context modeling process.
>
>   Hierarchical models perform explicit scale-by-scale transformations that naturally produce multi-scale latents, where each scale roughly corresponds to a specific frequency band. Finally, it can conduct scale-wise coding process by using 'upper-scale latent as a condition to assist in modeling lower-scale one'. Based on this, integrating the proposed intra-scale and inter-scale explicit regularization into this process, essentially corresponds scale-by-scale to the entire pipeline of model feature transformation, latent modeling, and decoding reconstruction, making it easier to address issues such as spectral energy dissipation within scales and spectral aliasing between scales;
>
>   By contrast, for single-scale models with various context modeling, they rely on powerful conditional modeling capabilities to directly fit the single-scale latent conditional distribution. Essentially, it is still a conditional probability modeling of the single-scale latent obtained from a single-scale transformation (HPCM's hierarchical design is also a hierarchical conditional probability modeling of the single-scale latent). There is no explicit multi-scale transformation of the input signal corresponding to the multi-scale latent design, so it cannot guarantee the natural frequency decomposition process. In other words, in the context modeling process, it is difficult to ensure that different slice components of a single-scale latent can be effectively decomposed on the spectrum and modeled scale by scale. Blindly using such regularizations may even disrupt the original context modeling design.
>
> - **Quantitative results:** Our experimental results on ELIC, MLIC++, and HPCM have also validated our analysis. The results are shown in the table below:
>
> |            |                                            | w/o regu | w/ regu |
> | ---------- | ------------------------------------------ | -------- | ------- |
> | ELIC       | BD-Rate over VTM-22.0 on Kodak dataset (%) | -3.56    | -3.88   |
> |            | Numbers of Epochs                          | 220      | 205     |
> | MLIC++     | BD-Rate over VTM-22.0 on Kodak dataset (%) | -9.22    | -2.56   |
> |            | Numbers of Epochs                          | 525      | 490     |
> | HPCM-Large | BD-Rate over VTM-22.0 on Kodak dataset (%) | -15.55   | -7.67   |
> |            | Numbers of Epochs                          | 475      | 490     |
>
> It can be seen that integrating the proposed regularization on ELIC can bring slight performance and training speed gains, but it is not as obvious as our hierarchical structure. In the context modeling processes of MLIC++and HPCM, which are more complex, performance degradation even occurs. This suggests that additional customized designs may be needed to adapt to single-scale VAEs, which is also a promising direction for our future work.
>
> **Refs:**
>
> Thanks again for the thoughtful review. We hope the above clarifications address your concerns and would be grateful if you consider raising your score.
>
> [1] Sønderby C K, Raiko T, Maaløe L, et al. Ladder variational autoencoders[J]. Advances in neural information processing systems, 2016, 29.
>
> [2] Vahdat A, Kautz J. NVAE: A deep hierarchical variational autoencoder[J]. Advances in neural information processing systems, 2020, 33: 19667-19679.
>
> [3] Duan Z, Lu M, Ma J, et al. Qarv: Quantization-aware resnet vae for lossy image compression[J]. IEEE Transactions on Pattern Analysis and Machine Intelligence, 2023, 46(1): 436-450.
>
> [4] Mallat S G. A theory for multiresolution signal decomposition: the wavelet representation[J]. IEEE transactions on pattern analysis and machine intelligence, 2002, 11(7): 674-693.
>
> [5] Xu Z Q J, Zhang Y, Luo T, et al. Frequency principle: Fourier analysis sheds light on deep neural networks[J]. arXiv preprint arXiv:1901.06523, 2019.

---

### Official Review · Reviewer_NjpX · 2025-10-31

**Soundness:** 3
**Presentation:** 3
**Contribution:** 3
**Rating:** 6
**Confidence:** 5

**Summary:**

This paper investigates the optimization challenges in hierarchical learned image compression via spectral analysis. The authors identify two major issues—cross-scale energy dispersion and spectral aliasing—and introduce two regularization strategies to mitigate them: (1) intra-scale frequency regularization through progressive DCT truncation, and (2) inter-scale latent regularization using similarity penalties. Experimental results demonstrate strong performance, achieving a 20.65% BD-Rate improvement over VTM-22.0 and a 2.3× training speedup without adding inference complexity.

**Strengths:**

1. The spectral interpretation of hierarchical training dynamics provides an intuitive understanding of cross-scale interactions and training instability.
2. Strong empirical results: 1) State-of-the-art compression performance across multiple datasets, 2) Significant 2.3× training acceleration without inference overhead, 3) Robust performance across diverse resolutions (480p–4K).
3. The proposed regularizers are training-only and do not increase inference complexity, making the method easy to deploy.
4. The identification of spectral dispersion and aliasing as key bottlenecks adds valuable understanding to hierarchical compression models.

**Weaknesses:**

1. The inter-scale regularization term minimizes $ L2(z_{l-1}, Conv(DWT(z_l)))$, which encourages similarity between adjacent scales. However, the text claims the objective is to make features “as distant as possible”. This appears contradictory and should be clarified.
The paper lacks a formal explanation of how minimizing latent similarity mitigates spectral aliasing. The relationship between spatial-domain similarity and frequency-domain decoupling needs stronger theoretical grounding.
2. The model diagram includes the FSP component, but it is not sufficiently discussed in the text. Compared to QARV, it remains unclear how much FSP contributes to convergence speed and rate–distortion performance. FSP seems conceptually close to AuxT [1], yet no detailed comparison or ablation is provided, despite citing AuxT.
3. The reparameterized block structure is mentioned but not thoroughly evaluated and its impact on performance should be quantified.
4. Hierarchical VAE has been previously applied to image compression (e.g., by Yueyu Hu et al. [2-3] ); this line of work should be properly cited and discussed.
5. It would be valuable to analyze whether the proposed regularization strategies could be extended to models such as HPCM or MLIC++, where complex context modeling divides single-scale features into multiple slices. Would regularizing these slices be similarly effective?

[1] Li, Han, et al. "On disentangled training for nonlinear transform in learned image compression." arXiv preprint arXiv:2501.13751 (2025).

[2] Hu, Yueyu, et al. "Learning end-to-end lossy image compression: A benchmark." IEEE Transactions on Pattern Analysis and Machine Intelligence 44.8 (2021): 4194-4211.

[3] Hu, Yueyu, Wenhan Yang, and Jiaying Liu. "Coarse-to-fine hyper-prior modeling for learned image compression." Proceedings of the AAAI Conference on Artificial Intelligence. Vol. 34. No. 07. 2020.

**Questions:**

Please refer to Weaknesses.

---

> ### Author Response · Authors · 2025-11-20
> **Response to Reviewer NjpX Part 1**
>
> Thanks for your insightful review and statement of the “**state-of-the-art compression**,” “**robust performance**”, “**easy to deploy**”, and "**adds valuable understanding to hierarchical compression models**". Below is the detailed response to each question, hope you can find them helpful:
>
> **Q1: Further explanation of the inter-scale regularization**
>
> We clarify that the inter-scale regularization is designed to *minimize* latent similarity across scales. There is a small typo in Eq. (6): the regularization term should be preceded by a “−” (minus) sign rather than a “+”. We apologize for the ambiguity. We have corrected it in the revised manuscript and supplemented relative explanation in Sec. 3.3 and Appendix A.1.
>
> Next, we elaborate on the motivation and mechanism:
>
> - **Problem setting.** hierarchical coding  is designed to progressively refine representations across scales[1] [2]: the latent from a coarser scale is used as a condition to predict the next finer scale. Often the architecture even explicitly uses residual addition between scales (see Fig. 9 latent block). In image signals, such residuals commonly correspond to higher-frequency details, while shallower latents capture lower-frequency structure (consistent with classical predictive / wavelet coding phenomena[3]). However, simple one-shot synchronous optimization across all scales lacks an explicit mechanism that ensures this intended sequential residual learning, producing redundancy across scales and manifesting as the spectral aliasing observed in Fig. 1.
> - **Principle.** The inter-scale regularization aims to align the coarser-scale latent to the subspace of the finer-scale latent (via a frequency-aware transform) and then penalize predictable (i.e., similar) components. Intuitively, if a component of a finer-scale latent can be well predicted from the previous scale, it is likely a low-frequency component that can be better represented at the coarser scale. Penalizing this predictable similarity removes those low-frequency components from the finer latent and leaves it to capture higher-frequency, less predictable residuals—thereby mitigating cross-scale spectral aliasing.
> - **Implementation.** We perform an explicit frequency-aware alignment by applying a discrete wavelet (DWT)-based transform to the coarser latent, which decomposes it into frequency sub-bands. We then apply a 1×1 convolution across channels to linearly map and recompose frequency sub-bands so they align with the finer-scale latent’s frequency channels. The inter-scale penalty then discourages the aligned coarser latent from predicting the same low-frequency content in the finer latent. Thus, the finer-scale latent retains the difficult-to-predict high-frequency components while redundant low-frequency parts are suppressed, alleviating spectral aliasing.
>
> **Q2: FSP module’s role and necessity**
>
> Thanks for this insightful question. Our main contribution focus on the intra-scale and inter-scale regularization, while the FSP module was only used as a skip-like component between scales in our hierarchical coding  architecture. In practice, we added FSP only during a finetuning stage after the main regularization training, and observed ≈0.68% bitrate savings. However, further experiments revealed that this gain primarily resulted from more complete finetuning under an adjusted LR scheduler setup (ReduceLROnPlateau patience increased from 2 to 5), not from FSP itself. Equivalent finetuning schedules without FSP produced similar rate-distortion gains, and adding FSP did not produce noticeable training speedups. Therefore, FSP is not necessary for our hierarchical architecture. The ablation results are reported in Table 1 and supplemented in the revisited version Appendix.A.1 and A.3.
>
> |                                        | w/o FSP + unfull training (wrong setup) | w/ FSP + full training | w/o FSP + full training |
> | -------------------------------------- | --------------------------------------- | ---------------------- | ----------------------- |
> | BD-Rate over VTM-22.0 on Kodak dataset | -19.05                                  | -19.73                 | -19.7                   |
> | Numbers of Epoches in finetune stage   | 46                                      | 78                     | 74                      |
>
> Moreover, we emphasize that the introduction of FSP is only related to the performance of the basic model itself, and does not affect the contribution of regularization that we mainly propose in paper.

---

> ### Author Response · Authors · 2025-11-20
> **Response to Reviewer NjpX Part 2**
>
> **Q3: Re-parameterization design's role**
>
> We re-parameterize the BasicBlock’s original 3×3 depthwise conv into a three-branch structure (3×3, 1×1, and identity) during training; at test time these branches are fused into a single convolution. This training-time multi-branch design effectively expands model capacity. To quantify its effect, we trained a network without this re-parameterization and compared rate-distortion and training speed; the results are summarized in Table 2 and added into the revised manuscript Appendix.A.x. Besides, It is worth emphasizing that while re-parameterization yields some improvement, the regularization contributions reported in the paper are independent of this effect because baselines also used re-parameterization.
>
> |                                            | w/o Rep | w/ Rep | w/o Rep but keep similar KMACs |
> | ------------------------------------------ | ------- | ------ | ------------------------------ |
> | BD-Rate over VTM-22.0 on Kodak dataset (%) | -16.45  | -19.73 | -18.1                          |
> | Numbers of Epochs in Full Training         | 350     | 380    | 500                            |
>
> **Q4: Prior hierarchical VAE works**
>
> Thank you, we have added discussion and citations of the mentioned hierarchical VAE and related hierarchical coding  works in the revised manuscript.
>
> **Q5: Applicability to single-scale models with complex context modeling**
>
> This is a valuable suggestion. We had similar thoughts before and performed preliminary experiments integrating our regularization into ELIC[4], MLIC++[5], and HPCM[6]. For ELIC with spatial & channel-wise context modeling, it performs slight training acceleration and RD gain. But for larger models with more complex context (e.g., MLIC++ with multi-reference design, HPCM with hierarchical context design), integrating our regularization instead decreased final performance. We provide both qualitative analysis and quantitative results below.
>
> - **Qualitative analysis:** We first analyzed the essential differences between hierarchical coding  structures and single-scale ones with various context modeling process.
>
>   Hierarchical models perform explicit scale-by-scale transformations that naturally produce multi-scale latents, where each scale roughly corresponds to a specific frequency band. Finally, it can conduct scale-wise coding process by using 'upper-scale latent as a condition to assist in modeling lower-scale one'. Based on this, integrating the proposed intra-scale and inter-scale explicit regularization into this process, essentially corresponds scale-by-scale to the entire pipeline of model feature transformation, latent modeling, and decoding reconstruction, making it easier to address issues such as spectral energy dissipation within scales and spectral aliasing between scales;
>
>   By contrast, for single-scale models with various context modeling, they rely on powerful conditional modeling capabilities to directly fit the single-scale latent conditional distribution. Essentially, it is still a conditional probability modeling of the single-scale latent obtained from a single-scale transformation (HPCM's hierarchical design is also a hierarchical conditional probability modeling of the single-scale latent). There is no explicit multi-scale transformation of the input signal corresponding to the multi-scale latent design, so it cannot guarantee the natural frequency decomposition process. In other words, in the context modeling process, it is difficult to ensure that different slice components of a single-scale latent can be effectively decomposed on the spectrum and modeled scale by scale. Blindly using such regularizations may even disrupt the original context modeling design.
>
> - **Quantitative results:** Our experimental results on ELIC, MLIC++, and HPCM have also validated our analysis. The results are shown in the table below:
>
> |            |                                            | w/o regu | w/ regu |
> | ---------- | ------------------------------------------ | -------- | ------- |
> | ELIC       | BD-Rate over VTM-22.0 on Kodak dataset (%) | -3.56    | -3.88   |
> |            | Numbers of Epochs                          | 220      | 205     |
> | MLIC++     | BD-Rate over VTM-22.0 on Kodak dataset (%) | -9.22    | -2.56   |
> |            | Numbers of Epochs                          | 525      | 490     |
> | HPCM-Large | BD-Rate over VTM-22.0 on Kodak dataset (%) | -15.55   | -7.67   |
> |            | Numbers of Epochs                          | 475      | 490     |
>
> It can be seen that integrating the proposed regularization on ELIC can bring slight performance and training speed gains, but it is not as obvious as our hierarchical model. In the context modeling processes of MLIC++and HPCM, which are more complex, performance degradation even occurs. This suggests that additional customized designs may be needed to adapt to single-scale VAEs, which is also a promising direction for our future work.

---

> ### Author Response · Authors · 2025-11-20
> **Response to Reviewer NjpX Part 3**
>
> Thanks again for your careful and responsible review. We hope these clarifications address your concerns, and we would be grateful if you could consider increasing your score.
>
> **Refs:**
>
> [1] Sønderby C K, Raiko T, Maaløe L, et al. Ladder variational autoencoders[J]. Advances in neural information processing systems, 2016, 29.
>
> [2] Vahdat A, Kautz J. NVAE: A deep hierarchical variational autoencoder[J]. Advances in neural information processing systems, 2020, 33: 19667-19679.
>
> [3] Mallat S. A wavelet tour of signal processing[M]. Elsevier, 1999.
>
> [4] He D, Yang Z, Peng W, et al. Elic: Efficient learned image compression with unevenly grouped space-channel contextual adaptive coding[C]//Proceedings of the IEEE/CVF conference on computer vision and pattern recognition. 2022: 5718-5727.
>
> [5] Jiang W, Yang J, Zhai Y, et al. Mlic++: Linear complexity multi-reference entropy modeling for learned image compression[J]. arXiv preprint arXiv:2307.15421, 2023.
>
> [6] Li Y, Zhang H, Li L, et al. Learned image compression with hierarchical progressive context modeling[C]//Proceedings of the IEEE/CVF International Conference on Computer Vision. 2025: 18834-18843.

---

> ### Comment · Reviewer_NjpX · 2025-11-20
> **Response to Authors**
>
> Thank you for your response. I am satisfied with the reply.

---

> > ### Author Response · Authors · 2025-11-20
> >
> > Thanks again for your valuable time and efforts in enhancing our manuscript.

---

### Author Response · Authors · 2025-11-20
**Overall response to all reviewers**

We thank all reviewers for their constructive reviews and recognition of our spectral-analysis idea (**NjpX,QW9U,HdPe,RfuU**), state-of-the-art performance (**NjpX,QW9U,RfuU**), and sufficient experiments (**NjpX,QW9U,HdPe,RfuU**). In response to their comments, we have added relative explanations, experiments, and analyses, and updated the revisited manuscript accordingly (**changes are marked in blue**). Concretely, we have:

1. **Provided further explanation of our method’s mechanism and justification**, especially clarifying how the inter-scale regularization alleviates spectral aliasing (Sec. 3.3 and Appendix A.1).
2. **Added the suggested ablations and analyses**. We appreciate that these suggestions are insightful and believe they can assist in further improving clarity. Also, we emphasize that our manuscript focuses on spectral analysis and regularized training for hierarchical coding  structures; the added ablations are targeted adjustments around the network structure and do not change our main contributions and conclusions.

   a. **Clarified the role and necessity of the FSP module**. FSP acts like a skip-connection module between different scales in our hierarchical coding  design, which intends to aid training. However, it is not intrinsic to the proposed regularization. Additional experiments show that the small BD-rate gain (≈0.68%) attributable to FSP originates from more thorough finetuning under a correct learning-rate scheduler setup; removing FSP has negligible effect on final performance and training speed. Hence, we will add another version of network design without the FSP module as reference in the revisited version. After peer review, we will then update relative results in the final paper.

   b. **Quantified the effect of the re-parameterized block design**. We emphasize that the baseline networks (without our spectral regularization) also used re-parameterization. This is related to the design of the basic model, which provides good basic performance. Our proposed regularization method further achieves performance improvement.
3. **Explored integrating our regularization into single-scale architectures with various context modeling (e.g., ELIC, MLIC++, HPCM)**. Although our regularization schemes are designed for hierarchical architectures, we still included preliminary experiments to examine applicability to single-scale designs. From both theory and experiments we explain why naively applying our regularization to single-scale models is inappropriate: unlike hierarchical models—where multiscale latents naturally emerge from input transformations—single-scale models produce a single-scale latent that lacks explicit frequency separation. Without architectural guidance or frequency-aware constraints, the single-scale latent cannot be reliably decomposed into multi-frequency components, and blindly introducing our regularizers may disrupt the original context-modeling design.
4. **Revisited and refined various statements, figures and formulas, and corrected several minor typos to improve readability.**

Additional explanations and results have been incorporated in the responses to each reviewers. After peer review, these revisions and results will be incorporated into the final paper.

---

### Comment · Area_Chair_FFLU · 2025-11-27

Dear reviewers,

Please review the rebuttal and discuss with the authors if you have not done it.

Thanks,
AC

---

### Author Response · Authors · 2025-11-30
**Explanation from the authors**

Dear Program Chairs, Senior Area Chairs and Area Chairs:

This is a statement from the authors regarding the recent leakage issue. We hope everyone is aware of it.

We understand the unexpected situation, and **we fully respect and comply with the double-blind policy of ICLR and OpenReview throughout the entire review and rebuttal process**. In addition, we would like to clarify that the previous score increase was due to **the reviewers’ recognition of our further explanations and improvements in the rebuttal, which effectively addressed their concerns on November 23rd**. We have full confidence in the professionalism of the ACs and understand that everyone is devoted to this work, to the conference, and to the development of the broader community. We sincerely hope that the ACs can understand our situation and provide a reasonable rating.

Thanks sincerely.

---

### Meta-Review · Area_Chair_ifTA · 2026-01-06

**Summary:**

The hierarchical coding approach is well established in image coding. This paper tackles optimizing hierarchical learned image compression using spectral analysis. They pinpoint two main bottlenecks: cross-scale energy dispersion and spectral aliasing. To address these during training, they propose two strategies: (i) intra-scale frequency regularization and (ii) inter-scale similarity regularization. Together, these yield notable gains: about a 20.65% average BD-Rate improvement over VTM-22.0 and a 2.3× faster training speed, while consistently beating both hierarchical and single-scale learned compression baselines across multiple datasets and resolutions. The method requires only training and adds no inference-time overhead, which is a clear strength. In the rebuttal, the authors respond to concerns in a respectful, constructive way. Overall, the final submission offers a solid contribution to learned image compression.

**Reviewer Concerns:**

concerns addressed in the rebuttal:

1. reviewers mainly questioned the theoretical clarity of the regularization terms, the role of auxiliary modules, and the stability of the method. overall, the authors gave a solid and detailed rebuttal that cleared up most points.
2. For the inter-scale regularization, several reviewers were confused by Eq. (6). It seemed contradictory to minimize similarity while reducing aliasing. The authors caught a sign typo and clarified that penalizing predictable low-frequency content at finer scales actually pushes them to model high-frequency residuals, reducing aliasing.
3. on the theoretical side, the original draft was thin, but the rebuttal improved it a lot. The authors linked hierarchical VAEs and spectral decomposition to the frequency principle and spectral bias, adding derivations, references, and clearer intuition. reviewers found this convincing, even if full formal proofs remain open for future work.
4. regarding auxiliary modules (FSP, re-param), the authors showed these aren’t central to the main method. ablations confirmed FSP has little impact once training is controlled, and they quantified the re-param effect. this helped clarify what really drives the gains.

a few reviewers still pointed out that the theoretical link between latent similarity penalties and strict spectral orthogonality could be tighter. the authors acknowledged this as future work, but it didn’t hurt the overall empirical strength of the paper.

**Reviewer Scores:**

This paper got positive scores at the first round of review, except for Reviewer RfuU. But Reviewer RfuU explicitly said that the stability of inter-scale regularization can be convincing. Thus, I would say all reviewers reach a consensus that this paper passes the acceptance borderline over 6.

---

### Decision · Program_Chairs · 2026-01-26

Accept (Poster)